# Calm ocean, stormy sea: Atmospheric and oceanographic observations of the Atlantic during the ARC ship campaign

Laura Köhler[1,2], Julia Windmiller[1], Dariusz Baranowski[3], Michał Brennek[3], Michał Ciuryło[3], Lennéa Hayo[4], Daniel Kępski[3], Stefan Kinne[5], Beata Latos[3], Bertrand Lobo[6], Tobias Marke[4], Timo Nischik[6], Daria Paul[4], Piet Stammes[7], Artur Szkop[3], and Olaf Tuinder[7]

[1]Max Planck Institute for Meteorology, Hamburg, Germany
[2]Alfred Wegener Institute, Bremerhaven, Germany
[3]Institute of Geophysics, Polish Academy of Sciences, Warsaw, Poland
[4]Institute for Geophysics and Meteorology, University of Cologne, Cologne, Germany
[5]Max Planck Institute for Chemistry, Mainz, Germany
[6]Department of Hydrography and Geodesy, HafenCity University Hamburg, Hamburg, Germany
[7]Royal Netherlands Meteorological Institute (KNMI), De Bilt, The Netherlands

**Correspondence:** Laura Köhler (laura.koehler@awi.de)

**Abstract.** During the Atlantic References and Convection (ARC) ship campaign with the reference MSM114/2, which took place in early 2023, the German research vessel Maria S. Merian travelled from Mindelo, Cape Verde, to Punta Arenas, Chile. One of the main objectives of ARC was to obtain vertically resolved cross sections of the Intertropical Convergence Zone (ITCZ). To this end, we crossed the ITCZ three times in the meridional direction. We present the atmospheric and oceanographic measurements collected during ARC in a standardized way to facilitate working with data from different instruments and to make the data easily accessible. This approach is not limited to ARC but could serve as a prototype for future (and past) ship campaigns. We present data from the integrated ship sensors (DShip), a Humidity and Temperature Profiler (HATPRO), a Ceilometer, aerosol instruments (DustTrak, Microtops, and Calitoo), radiosondes, Uncrewed Aircaft Vehicles (UAV), and Conductivity, Temperature, and Depth (CTD) profiles of the upper ocean. We distinguish temporal continuous data (DShip, HATPRO, Ceilometer, DustTrak) from point measurements (radiosondes, UAVs, CTDs, Calitoo, Microtops). To illustrate the data sets provided, we present examples of measurements taken during the three crossings of the ITCZ and during a storm in the Roaring Forties in the South Atlantic at the end of the campaign. An overview of all available data sets, including dois and download links, can be found in Köhler et al. (2024a) with the doi https://doi.pangaea.de/10.1594/PANGAEA.966616. For references to the individual data sets, please refer to the data availability section.

## 1 Introduction

The Atlantic Ocean and its interaction with the atmosphere is a major driver of global (Omrani et al., 2022) and local (Yin and Zhao, 2021) weather and climate. Over the tropical Atlantic, the Intertropical Convergence Zone (ITCZ) plays a central role, not only defining the main region of precipitation over the ocean, but also strongly influencing the distribution of precipitation over neighbouring continents, with profound influences on life and economy, as exemplified by changes over the

Sahel (Held et al., 2005; Biasutti and Giannini, 2006). Despite its importance and seemingly large scale, the Atlantic ITCZ is poorly represented in models, with large and long-standing biases in its position (e.g. Richter and Xie, 2008; Siongco et al., 2017). Recently, the question has been raised whether an improvement in the large-scale representation of the ITCZ requires a better representation of the processes shaping the mesoscale structure of the ITCZ (Klocke et al., 2017; Masunaga, 2023; Windmiller and Stevens, 2024). To answer this question, observational data are of key importance, serving as a benchmark for both numerical and theoretical models. While satellite observations provide invaluable information covering extended periods and areas, a major shortcoming of studying the ITCZ based on satellite data alone is the still limited vertical resolution and availability of data below clouds. One important source of measurements with high vertical resolution over oceans are ship-based measurement campaigns. In this paper we present data collected in early 2023 during the Atlantic References and Convection (ARC) campaign aboard the German research vessel (RV) Maria S. Merian. The ARC reference at the German Research Fleet Coordination Centre is MSM114/2 (Nitsche, 2023).

ARC took place from 22 January to 23 February 2023. During this period, the RV Maria S. Merian travelled from Mindelo, Cape Verde to Punta Arenas, Chile, see Figure 1(a). The three main goals of ARC were to obtain vertically resolved cross sections of the ITCZ, to collect atmospheric and hydrographic reference data, in particular for comparisons with the Aeolus satellite and to contribute to the Seabed 2030 database, a global initiative to obtain a complete seabed map by the end of the decade (Mayer et al., 2018), and to examine the protist populations in the Atlantic along the route. During ARC, atmospheric and oceanographic data were collected by a number of different instruments. These instruments can be divided into those that operated continuously and those that only took measurements at specific times. The continuous measurements during ARC include the ship-integrated instruments of the DavisShip system (DShip), the Humidity And Temperature PROfiler (HATPRO), the Ceilometer, and the DustTrak. The DShip data we include here contain weather and surface water observations from the Weather Station and Pure Sea Water System respectively, as well as radiation, wave properties from the Wave Monitoring System, and ship data such as position from the navigation system. Point measurements include instruments with vertical resolution, in particular, radiosondes, Uncrewed Aircraft Vehicle (UAV) measurements and Conductivity, Temperature and Depth (CTD) profiling of the ocean, as well as vertically integrated measurements from sun photometers.

The data collected by the instruments listed above are originally provided in different file formats, require different processing which, in turn, requires specific knowledge of the measurement principle, use different conventions, are owned by different institutions, and are provided at different temporal and/or spatial resolutions. This makes it difficult to use the collected data in a holistic way, especially if the data are also published in different places. As this challenge is not unique to the ARC campaign, the purpose of this paper is twofold. First, to publish a coherent dataset of the data collected during the campaign which facilitates studying research questions using different variables from different instruments as well as comparisons of the same quantity measured by different instruments with different techniques. Second, to provide the necessary processing steps and scripts to serve as a prototype for future (and past) ship campaigns. In particular, we propose a standardised way of making data from different instruments available, using ARC as an example. Data from all instruments are provided in the same file format with standardised variable names and are published jointly so that all instruments can be accessed at the same place

(Köhler et al., 2024a). The publication of the data is accompanied by a GitHub repository (Köhler, 2023) containing the exact
settings for the processing, for which we have developed the Python package shipspy (Köhler, 2024).

ARC took place during boreal winter and therefore complements the Mooring Rescue ship campaign in summer 2021 with
the RV Sonne, reference SO284 (Brandt et al., 2021), and the BOW-TIE ship campaign with the RV Meteor, reference M203, in
summer 2024 (Leitstelle Deutsche Forschungsschiffe, 2024a). Although they were conducted at different times of the year and
investigated different parts of the tropical Atlantic than the ARC campaign, a common core objective of the campaigns was to
create vertical profiles of the ITCZ through the thermocline to the tropopause. An additional ship campaigns with a similar suite
of instruments is planned for the beginning of 2025 (M207 with RV Meteor (Leitstelle Deutsche Forschungsschiffe, 2024b)).
Processing the data from the various campaigns in the same way as for ARC will greatly facilitate the study of seasonal and
regional changes in the Atlantic ITCZ.

Section 2 introduces the research vessel Maria S. Merian and the ship's track during the ARC campaign. In section 3 we
introduce the instruments and describe the data processing. In section 4 we provide information on the data sets. Finally, to
give a first overview of the collected data, in section 5 we show some examples of observations for three meridional crossings
of the ITCZ, a storm in the Roaring Forties, as well as for the whole measurement period. We conclude with a summary in
section 6.

## 2   Research vessel and cruise track

The RV Maria S. Merian is a German research vessel built in 2005 especially for ice margin research. It has a total length of
94.76 m, a width of 19.2 m, a maximum total height of 38 m, and an unladen weight of 4493 t. The draught is about 6.5 m. The
scientific payload can be up to 150 t. During ARC, the mean ship speed was 9.8 kn. Additionally to the 24 crew members, the
Maria S. Merian can host up to 23 scientists. 15 scientists joined the ARC campaign. More information about the ice margin
research vessel Maria S. Merian can be found in the manual (von Bröckel et al., 2021).
The RV Maria S. Merian left Mindelo located at 24.98°W 16.88°N on 22 January. After leaving Mindelo, we travelled south
to obtain three cross sections of the ITCZ at about 23°W, see the top inset of figure 1(a). By choosing this longitude, we were
able to collect data near the PIRATA buoys (Bourlès et al., 2019) at 23°W and 4°N as well as at the equator. This enables
a comparison between the data collected during ARC and the long-term measurements of the buoys. The exact longitude of
the ship track was shifted by about 0.15° for each crossing so that new bathymetric data could be collected, contributing to
Seabed 2030 through seafloor cartography. The track in the inset is thus colour coded indicating the time starting with blue on
26 January via red to green on 5 February. In general, underway echo soundings were performed during most of the campaign.

For each cross-section, the latitudes of the ship's turning points were chosen so that we obtained a complete meridional
transect of the ITCZ. Since the synoptic variability in the position of the ITCZ is as large or larger than the width of the ITCZ
(Chen and Ogura, 1982; Frank, 1983), we relied on both on-board measurements and satellite images to determine the latitudes
of the turning points at short notice. Fig. 2 shows an example of a GOES 16 satellite image of the Atlantic ITCZ on 4 February
2023. While the ITCZ can be defined in different ways, e.g. based on occurance of deep convective clouds, surface wind speed

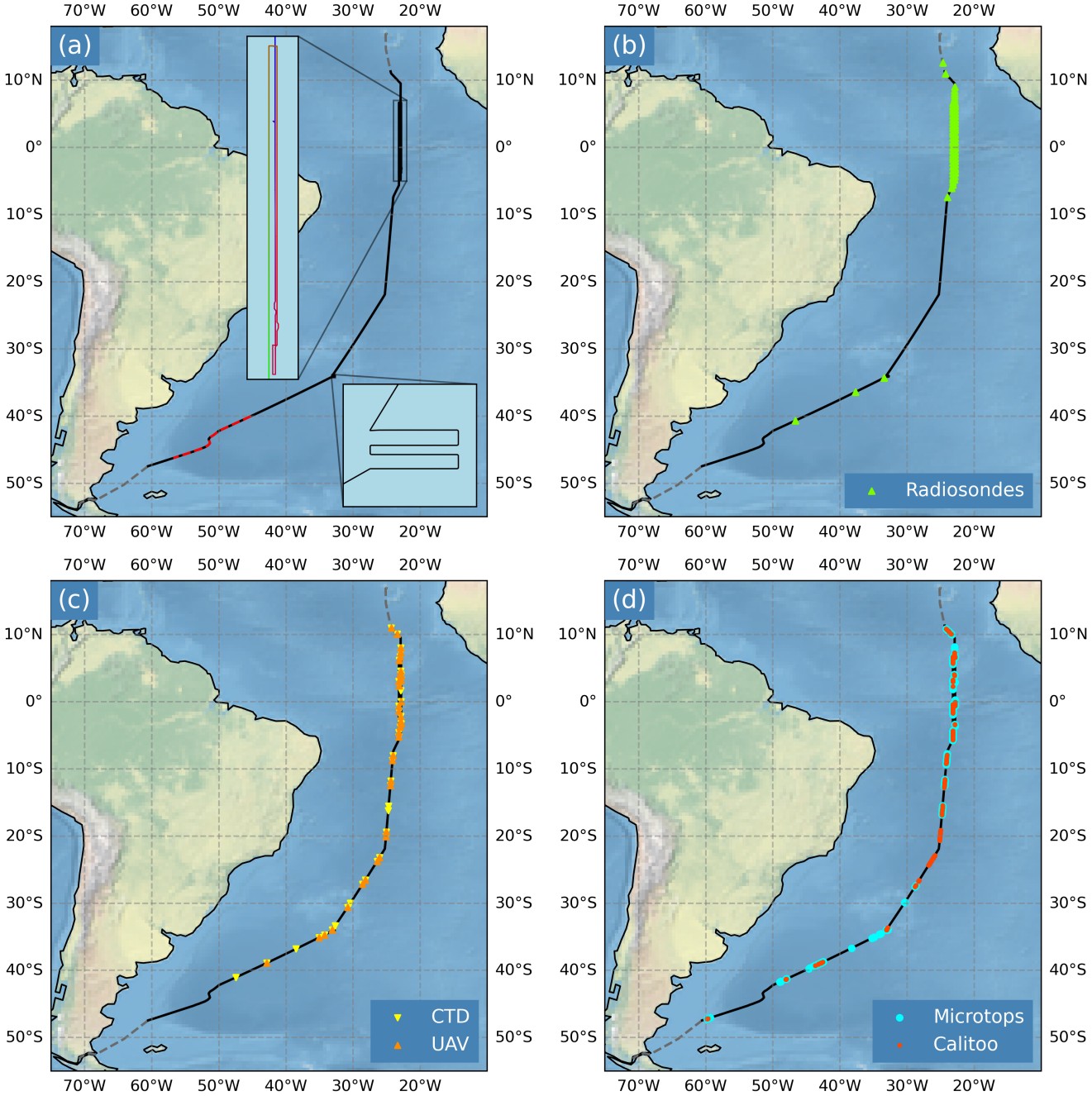

**Figure 1.** Ship track of the ARC campaign and positions of point measurements. The black line denotes the region when the ship was in international waters, the parts of the track in national waters are sketched in gray. (a) Route with zooms in the insets. The colour coding in the upper inset refers to the time from blue (26 Jan) via red to green (5 Feb). The red dashed line indicates the position of the storm. (b) Positions of radiosonde launches (green, see section 3.7), (c) CTD (yellow, see section 3.9) and UAV (orange, see section 3.8) positions, and (d) Calitoo (red, see section 3.5) and Microtops (blue, see section 3.6) positions.

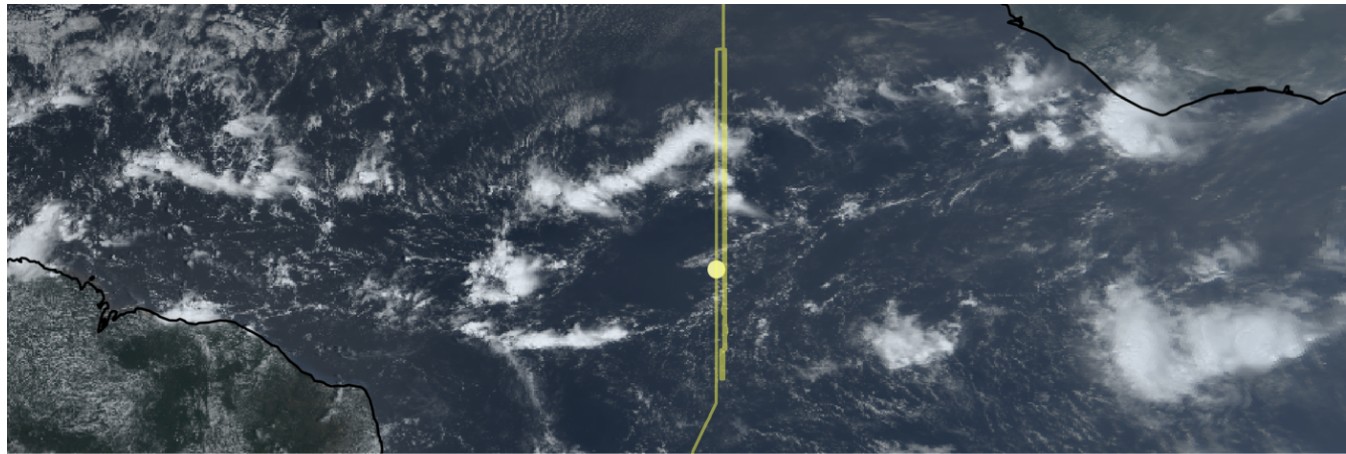

**Figure 2.** Image of the Atlantic ITCZ with a pronounced edge intensification (cf. Masunaga (2023)) from GOES 16 satellite on 4 February, 2023 at 12:45 UTC. The yellow dot denotes the position of the RV Maria S. Merian. The yellow line indicates the ship track for reference. Credit: NOAA Geostationary Operational Environmental Satellites (GOES) 16, 17 & 18 was accessed on 2023-10-27 from https://registry. opendata.aws/noaa-goes.

or precipitation rate, we based our decisions on the column water vapour (CWV) field. In particular, we used the moist margins, i.e. CWV equal to 48 mm, to decide on the turning points as Mapes et al. (2018) shows that the deep convection of the ITCZ is limited to within these margins. After leaving the Atlantic ITCZ, the RV Maria S. Merian headed for its port of disembarkation in Punta Arenas (70.91°W 53.15°S), Chile. At 33°W 34°S the ship track was modified to map a sea mount as shown in the lower inset of Figure 1(a).

## 3 Instrumentation and processing

In this section, we present the atmospheric and oceanographic instruments and comment on the post-processing required. The variable name in the final data set is given in brackets when it differs from the name mentioned in the text. Table 1 gives an overview of the instruments and their included sensors as well as some additional information such as the mounting position on the ship. The instrument positions are also shown in Figure 3, where the places are marked in a picture of the RV Maria S. Merian taken with the UAV on 26 January, 2023. Most of the instruments were connected to the ship network and thus their time stamps were automatically synchronised with DShip or used GPS timestamps which were in agreement with DShip. For the DustTrak, the time was manually synchronised with DShip on a regular basis. All time stamps were checked during the post processing.

Most instruments were either not sensitive to the ship's motion or they were adapted to it in various ways. The radiation instruments on the RV Maria S. Merian are mounted on a two-axes gimbal for horizontal stability. The sunphotometers were handheld and thus adapted to the ship's motion by manually pointing to the sun. The DustTrak was not sensitive to the ship's

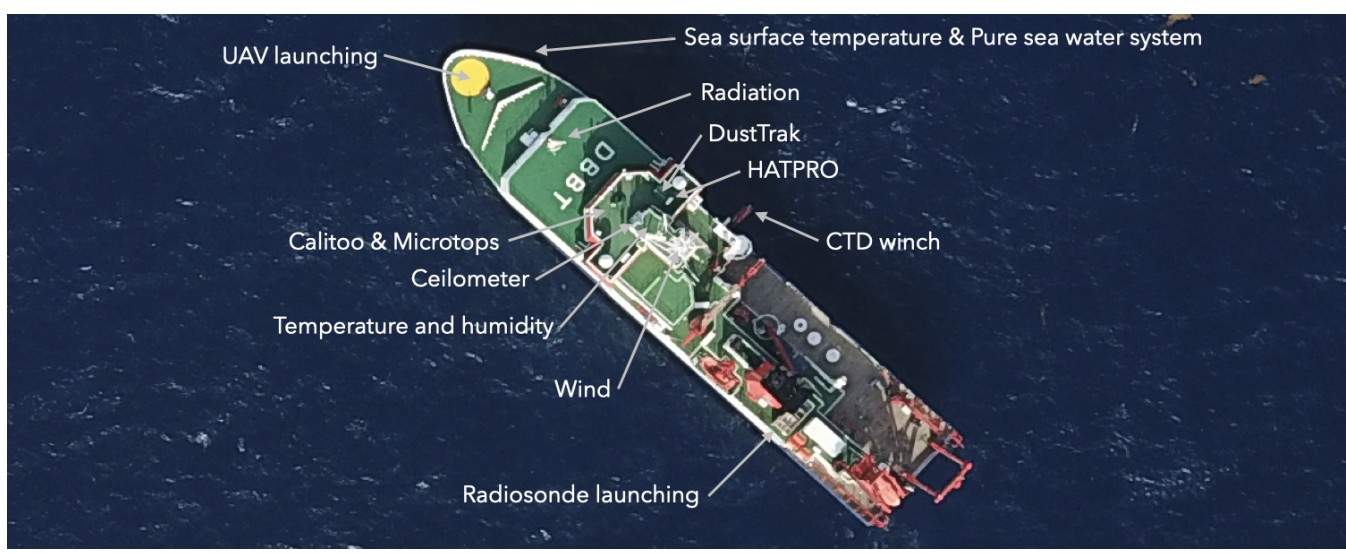

**Figure 3.** Locations of the instruments and sensors on the RV Maria S. Merian which has a length of 95 m and a width of 19 m. The picture of the ship was taken on 25 January at 16:36 UTC with the UAV at a height of 513 m.

motion because it pumps ambient air. The Ceilometer and the HATPRO were fixed on the observation deck, so they moved
with the ship. However, we do not consider this a significant error since the angular deviation was small ($< 6.2°$) except during the storm ($< 22.2°$) where also strong precipitation and salt water from large waves were deteriorating the measurements. The information about the ship's motion is given in the DShip data set parameters ship_roll and ship_pitch.

In addition to the instruments listed below, we had a cloud camera installed on the upper observation deck (Kinne and Köhler, 2024a). In the published data set, we include hourly all sky snapshots to estimate cloud cover and cloud type. Note that
we do not include the biological observations or the bathymetry data as these will be published elsewhere.

Table 1: Overview of sensors in the different instruments and the device position on the ship.

| Instrument | Sensor name | Comment |
|---|---|---|
| DavisShip system (DShip) | Werum Software & System AG | |
|   Weatherstation | EUCAWS | |
| |   Vaisala PTB | yellow deck (10.5 m) |
| |   Lufft Ventus ultra sound sensor | main mast (33.5 m) |
| |   E+E EE33 | upper observation deck (21.5 m) |
| |   PT100, 1/3 DIN B | intake point: 2 m |
|   Pure sea water system | Sea-Bird SBE45 | intake point depth: 6.2 - 6.8 m |
| | WETLabs FLNTUS | |
|   Wave monitoring system | WaMoS II | with Rutter Radar 100S6 |

| Instrument | Sensor name | Comment |
|---|---|---|
| Radiation<br><br>Navigation system | Kipp & Zonen CMP 21 & CG 1<br>Kongsberg SeaTex AS Seapath 320<br>fiber optic gyro compass<br>Kongsberg (Simrad) EM 122 | front mast (20 m) |
| HATPRO | RPG-HATPRO G5 | observation deck (16 m) |
| Ceilometer | Lufft CMH 15K) | upper observation deck (18 m), Nd:YAG Laser |
| DustTrak | DRX 8533, TSI Inc. | observation deck (16 m) |
| Calitoo sun photometer | Tenum Calitoo | observation deck (16 m) |
| Microtops sun photometer | Dpro Scientific model 540 Microtops II | observation deck (16 m), part of Maritime Aerosol Network |
| Radiosonde | Vaisala DigiCORA Sounding System MW 41<br>sounding processing system SPS311G<br>ground check subsystem RI41<br>portable antenna set CG31<br>radiosonde Vaisala RS41-SGP | |
| Uncrewed Aerial Vehicle (UAV) | DJI Mavic PRO<br><br>InterMet iMet-XQ2<br>Sparvio SKH1<br>LI-COR LI-550 TriSonicaTM Mini<br><br>Alphasense OPC-N3<br><br><br>RBR Solo3T | Atmospheric profiles up to 500 m<br>mounted on top, T/rh/p<br>mounted on top, T/rh/p<br>mounted on a 15 cm mast on top, u,v<br>mounted on top with 30cm-long intake chimney, pm1, pm2d5, pm10<br>tethered on 20 m line underneath, upper ocean T |
| Conductivity, Temperature, and Depth (CTD) | Sea-Bird SBE 911plus, SN-0807<br>Sea-Bird SBE3, SBE 4 & SBE 43<br>WETLabs ECO FLNTUR(TR)D<br>Digiquartz 0807<br>Sea-Bird QSP 2300<br>Benthos PSA-916 | |

## 3.1 DavisShip system

The DavisShip system (DShip) from Werum Software & System AG is used to collect, combine, and store data from the sensors permanently installed on the ship. In total, it contains data from about 250 sensors, including meteorological, oceanographic, and ship specific measurements. Most of the DShip instruments measure continuously and output is available in 1 s time steps. Here, we include data from the weather station, the pure seawater system, the wave monitoring system, the global radiation meter, and the navigation system. All outputs are averaged to a 1 minute temporal resolution and fill values are replaced by "not a number" (Köhler and Windmiller, 2024).

### Weather station

The automatic board weather station of type EUCAWS provides continuous meteorological measurements at a frequency of 1 Hz . Pressure is measured on the yellow deck at a height of 10.5 m by a Vaisala PTB sensor with a resolution of 0.1 hPa. The wind sensor, a Lufft Ventus ultra sound sensor, is installed on the main mast at a height of 33.5 m. It measures relative wind speed and direction with a resolution of $0.1 \text{ m s}^{-1}$ and $0.1°$, respectively. Temperature and relative humidity are measured with E+E EE33 sensors on the upper observation deck at a height of 21.5 m with accuracies of 0.1 K and 0.1%, respectively. Sea surface temperature is measured at 2 m depth by a PT100, 1/3 DIN B sensor with an accuracy of 0.1 K in the echosounder equipment room at the front tank deck starboard side. In the data set, we include bulb air temperature (t_air), pressure at sea level (p_air), wind speed (wspd), wind direction (wdir), relative humidity (rh), and sea surface temperature (sst_2m).

### Pure sea water system

The pure sea water system on board of the RV Maria S. Merian is designed to supply sea water to the ship's laboratories and measurement systems. The pure seawater is pumped into the ship. The seawater parameters are determined at the intake point. Temperature and conductivity are measured at a depth of 6.2 to 6.8 m with one of two SBE45 thermosalinograph sensors. Salinity is determined from the conductivity values. According to the manufacturer Sea-Bird, the sea surface temperature at the intake point (sst_7m) comes with an error of $\pm0.002$ K, conductivity with $\pm0.0003 \text{ S m}^{-1}$, and salinity with $\pm0.005 \text{‰}$. In addition, the pure seawater system measures chlorophyll A (chlorophyll_a) with a FLNTUS sensor from WetLabs which comes with an accuracy of $2.5 \cdot 10^{-8} \text{ kg m}^{-3}$.

### Wave Monitoring System

The Wave Monitoring System (WaMoS) II is a radar reanalysis system designed to provide real-time information on the sea state using microwave backscatter. It extracts the main parameters describing the sea state, such as wave amplitude, period, and direction from the temporal and spatial evolution of the backscatter profiles. On the RV Maria S. Merian, the WaMoS system is operated with a Rutter Radar 100S6. To determine the wave parameters, the data is sampled in image sequences and transformed into Cartesian coordinates, before applying a discrete Fourier transformation. To isolate the wave signal from the background noise, a bandpass filter matching the dispersion relation is applied. The two-dimensional spectrum is then

separated into different frequency directions so that the wave parameters can be extracted from the one-dimensional spectra. Finally, the wave parameters are averaged over a time interval of 1 minute.

WaMoS analyses the Fourier spectrum up to its second order. Here, we only include the leading order as higher orders do not give robust results. We include the sea surface mean wave period (wave_period), sea surface maximum wave length (wave_length), sea surface significant wave height (wave_height), sea surface wave direction (wave_dir), sea water speed (current_speed), and sea water direction (current_dir). The determination of the wave parameters, even to leading order, is not stable leading to large fluctuations in the values. Data is available for the whole campaign. The errors given by the manufacturer are 0.5 m or 10% for the wave height, 0.5 s for the wave period, 10% for the wave length, 2° for the main wave direction and the current direction, and 0.2 m s$^{-1}$ for the current speed. We have omitted unphysical values such as negative wave heights or wave periods from the data set. Note, however, that there are jumps in wave length and period of individual values that are significantly smaller than before and after. At these times, the Fourier analysis failed to determine the longest wave period.

**Global shortwave and longwave radiation**

Two instruments are used to measure atmospheric radiation on board: a pyranometer for shortwave irradiance (swr) in the broadband solar range, instrument type CMP 21 from Kipp & Zonen, and a pyrgeometer for longwave irradiance (lwr) in the broadband infrared range, instrument type CG 1 from Kipp & Zonen. Both instruments measure the radiation on a horizontal surface integrated over all downward directions, the so-called global radiation. The instruments are mounted on the front mast of the ship at a height of approximately 20 m above sea level. The instruments are mounted in a two-axes gimbal, such that their sensing surfaces are oriented horizontally. The data are measured at 1 Hz. The irradiances are given in W m$^{-2}$. The calibration of the radiation instruments was performed by the instrument manufacturer just before the cruise. The $3\sigma$ uncertainty is $\pm 7$ % for the shortwave radiation, and $\pm 6$ % for the longwave radiation.

**Navigation system**

Seapath is a navigation system that combines global navigation satellite systems and inertial measurements to determine the parameters needed to navigate the ship. The product installed is Seapath 320 from Kongsberg SeaTex AS. It uses two single frequency 12-channel GPS receivers for position and heading. The position of the ship (lat, lon) is given with an accuracy of $\pm 0.15$ m, the speed (ship_speed) is determined with an error of $\pm 0.07$ m s$^{-1}$, roll (ship_roll) and pitch (ship_pitch) come with an error of $\pm 0.03°$, and the heave (ship_heave) with $\pm 0.05$ m. For heave, pitch, and roll, we also provide the standard deviations (variable_std) for the 1-minute means since they vary on much smaller time scales than minutes. The measuring frequency can be up to 100 Hz. The heading (ship_heading) is measured by a fiber optic gyro compass with an accuracy of $\pm 0.7°$ secant latitude.

The depth (sea_floor_depth) is mostly measured with a Kongsberg (Simrad) EM 122 deep sea multibeam echo sounder. Only 0.4% of the data points are obtained by the parametric sediment echo sounder Atlas Parasound DS P-70. The error of the depth is $\pm 0.4$ m. Note that there are isolated data gaps in the depth data and we find outliers, which we did not remove, mainly sudden jumps to significantly smaller values. A reason for this could be fish passing under the ship or measurement errors.

## 3.2 HATPRO

The HATPRO (Humidity And Temperature PROfiler, Rose et al. (2005)) is a ground-based passive microwave radiometer, receiving downwelling radiation emitted by the atmosphere. It measures in seven channels in the K-band (22.24 - 31.4 GHz) and seven channels in the V-band (51.26 - 58.0 GHz). Emissions from water vapor, oxygen and liquid water are measured as brightness temperatures $T_B$ following Planck's law.

The HATPRO was installed on the observation deck of the ship at 16 m above sea level to avoid sea spray. An absolute calibration with liquid nitrogen was performed before the start of the campaign on 14 January, 2023 in Las Palmas. During the measurement period, the scan strategy consisted of continuous zenith measurements at 1 s temporal resolution and a boundary layer scan over 10 angles between $0°$ and $30°$ every 15 minutes to enhance the vertical resolution of the temperature profiles below 2 km. For the profiles the vertical resolution increases step wise with altitude from 50 m close to the ground to 500 m for heights above 5000 m. As a wet radome causes additional liquid emissions, contributing to the total column emissions, the data during precipitation events is flagged by utilizing the rain sensor of the attached weather station and removed in the post processing.

Retrievals trained with different input data were tested, and coefficients are chosen that were derived from a climatology of more than 10 000 radiosonde launches between 1990 to 2018 from Grantley Adams International Airport (GAIA, station ID 78954 TBPB) close to the Barbados Cloud Observatory. The retrieval method applied to the $T_B$ measurements, similar to Löhnert and Crewell (2003) and described in Schnitt et al. (2024), is based on a quadratic least square regression method to retrieve CWV, LWP, and absolute humidity profiles. A linear regression is used in case of the temperature retrieval. From that, further products are derived, including relative humidity, potential temperature, and equivalent potential temperature profiles. Retrieval error of CWV is estimated around 0.5-0.8 kg m$^{-2}$ (Steinke et al., 2015), and for LWP the relative uncertainty ranges from over 100 % for LWP below 15 g m$^{-2}$, to 50 % for LWP around 40 g m$^{-2}$, and decreases to 20 % for LWP above 100 g m$^{-2}$ (Jacob et al., 2019). Profiles of atmospheric temperature and humidity can generally be retrieved with errors below 2 K (Löhnert and Maier, 2012) and less than 1 g m$^{-3}$ (Walbröl et al., 2022), respectively, when compared to radiosondes in the lowest 4 km. This translates to an error of up to 0.2 in relative humidity. The information content decreases with height, with about 10 % and 20 % of the temperature and humidity information, respectively, coming from heights above 500 hPa (Ebell et al., 2013). Higher altitudes are therefore mostly influenced by the mean state given in the training data set.

Processing of the data is based on the Python package MWRpy (Marke et al., 2024), which has been developed for the Aerosol, Clouds and Trace Gases Research Infrastructure (ACTRIS, Laj and Coauthors (2024)) and implemented in the Cloudnet cloud classification scheme (Illingworth et al., 2007). The software reads in the raw data format of the instrument and produces files of quality controlled $T_B$ and retrieved quantities. Beside HATPRO quality flags, including rain detection and receiver thermal stability, also a spectral consistency check is provided. Therefore, $T_B$ for a specific channel is derived via statistical retrieval from other channels and compared to observed $T_B$, since the atmospheric information is not independent, and only certain atmospheric spectra are physically possible. With the application of a statistical LWP retrieval, nonphysical bias values can occur and be detected during clear sky cases. Therefore, an offset correction was applied to the retrieved LWP

using a 2 min brightness temperature standard deviation of the MWR window channel at 31.4 GHz. Liquid water cloud free
scenes are detected within a 20 min window using a threshold of 0.1 K times the median ratio of the water vapour (22.24 GHz)
and window channel to account for a water vapour dependency of the threshold. Offset values are also stored in the data set
(lwp_offset, cf. Table A1). The data is averaged to a 1 minute frequency. Unphysical values of the relative humidity larger than
1 or smaller than 0 were removed. We include data in Hayo et al. (2024) from the HATPRO until 15 February as data quality
could not be guaranteed for later times times due to the storm event and connected strong precipitation and salt deposits on the
instruments.

### 3.3 Ceilometer

The ceilometer is a LIDAR (Lufft CMH 15K) operating at 1064 nm using a Nd:YAG solid state laser with a pulse energy of
7 $\mu$J. The ceilometer was installed on the upper observation deck at a height of approximately 18 m and operated continuously
throughout the period in international waters. The backscattered signal can be used to derive cloud base height, cloud penetra-
tion depth and aerosol layer height of up to three layers in the vertical range between 5 m and 15 km. For cloud and aerosol
layer detection, the full backscatter profile is also stored in the level 0 raw data (Köhler et al., 2024f) but not integrated in the
processed data set (Köhler et al., 2024e). The field of view of the receiver is 0.45 mrad. The accuracy of the measured distances
is less than $\pm 5$ m or $\pm 0.2\%$.

For all quantities except the raw backscatter profiles (beta_raw and beta_raw_hr) we have omitted unphysical (negative)
values and taken one-minute averages. In addition to the height of the aerosol layer of the planetary boundary layer (pbl), we
include the total and base cloud cover (tcc and bcc, respectively), the sky condition index (sci), the vertical optical range, the
cloud base height and depth and their errors (vor, cbh_layer, cdp and var_error), the maximum detection height (mxd), the raw
signal (beta_raw and beta_raw_hr) and its standard deviation (std), and the base line (base). Note that we have included the
first detected cloud base height without and with time dependence (cbh and cbh_2s), retrieved as described in Nuijens et al.
(2014). The data set can be downloaded from Köhler et al. (2024e). For the technical quantities, we provide the level 0 raw
data as additional data set (Köhler et al., 2024f).

### 3.4 DustTrak aerosol monitor

The DustTrak aerosol monitor (desktop model DRX 8533, TSI Inc.) measures the mass of ambient aerosols per unit volume
in four size ranges: PM1 (pm1), PM2.5 (pm2d5), PM4 (pm4) and PM10 (pm10). Here PM$x$ means the particulate mass of
all particles with a diameter below $x$ micron, in units of kg m$^{-3}$. The total mass concentration of ambient aerosol particles
is included in the data by the variable pm_all. The principle of the DustTrak instrument is based on the measurement of light
scattered by the ambient aerosols which are pumped through the instrument (Wang et al., 2009). The instrument had been
calibrated by the supplier just before the campaign.

The DustTrak aerosol monitor was set-up on the observation deck of the ship, at 16 m above sea level. The instrument has a
shortest sampling rate of 1 s; we used 10 s integration time. The data were later averaged to 1 minute. We typically measured
semi-continuously during the daytime, excluding periods with rain, since the instrument was not rainproof. The data were

filtered by excluding measurements affected by the ship's chimney smoke, using the relative wind speed and direction of the airflow towards the instrument. Also other obvious influences from the ship in the data were removed.

Since the DustTrak measures PM$x$ by means of light scattering, its accuracy is affected by the shape, size, refractive index and density of the aerosol being sampled (Wang et al., 2009). We estimate the PM$x$ accuracy to be about $\pm 10$ %. The DustTrak data can be downloaded from Stammes et al. (2024b).

### 3.5 Calitoo sunphotometer

The Calitoo handheld sunphotometer manufactured by Tenum, France, measures the intensity of direct sunlight from which the aerosol optical thickness (AOT; also called aerosol optical depth, AOD) can be derived, assuming that there are no clouds. The instrument measures at three wavelengths: 465 nm (aot_465), 540 nm (aot_540), and 619 nm (aot_619), and has built-in GPS for date, time and geolocation (lat, lon). From the aerosol optical depths, we derived the Angstrom parameter (angstrom_exp). The data set also contains air temperature (t_air), pressure (p_air), and the solar elevation angle (elevation). Observations were performed on the observation deck of the ship, at 16 m above sea level (alt_sensor). The AOD values of the three Calitoo channels were offset-corrected such that all curves run through the origin when plotted against aot_465.

The user has to point the instrument towards the unobscured sun. The instrument stores the maximum radiance value and calculates the AOT, including a correction for extinction due to Rayleigh scattering and ozone absorption. These corrections were recalculated using the actual pressure and ozone data. We used two Calitoo instruments with serial numbers #108 and #109. The accuracy of the sunphotometer AOT is about $\pm 0.01$. Aerosol measurements with Calitoo published in Stammes et al. (2024a) were performed whenever possible as shown in Figure 1(d).

### 3.6 Microtops sunphotometer

The handheld Microtops sunphotometer measures the intensity of direct sunlight from which the AOT can be derived, in case there are no clouds. The Microtops on board determines AOTs in the 380 nm, 440 nm, 670 nm, and 870 nm channels. From the spectral slope in a double logarithmic representation, the Angstrom parameter is determined. The 940 nm channel, which is a water vapor absorption line, is used to determine the column water vapor under cloud free conditions. The Microtops instrument is part of the Maritime Aerosol Network (MAN, https://aeronet.gsfc.nasa.gov/new_web/maritime_aerosol_network_v3.html), which is the maritime extension of Aeronet (Smirnov et al., 2000, 2009). It was calibrated at NASA GSFC. Post processing, which includes filtering out remaining clouds, was done within MAN and the data is republished from the MAN website: https://aeronet.gsfc.nasa.gov/new_web/cruises_v3/Maria_Merian_23_0.html.

In the data set provided here (Kinne and Köhler, 2024b), we included the AOTs at 380, 440, 675, and 870 nm (aot_380, aot_440, aot_670, aot_870), and the interpolated value for 500 nm (aot_500), the Angstrom parameter (angstrom_exp), and the column water vapor (cwv). For all these quantities, standard deviations are provided as well (quantity_std). Additionally, the air mass coefficient (air_mass), the position (lat, lon), and the number of observations (number_obs) subsampled to one data point to minimize the error are provided. We used the series data downloaded from the MAN website. Furthermore, we removed

negative values of the Angstrom exponent. As for the Calitoo measurements, measurements with Microtops sunphotometer were performed whenever possible as shown in Figure 1(d).

## 3.7 Radiosondes

A total of 93 radiosondes were launched during ARC. Of these, 87 radiosondes were launched during the ITCZ crossings between 3 am on 26 January and 9 pm on 5 February, at a frequency of approximately every three hours. Two radiosondes were launched before entering the ITCZ and four radiosondes were launched on the way to Punta Arenas after the crossings. The radiosonde launch positions are indicated by the green triangles in figure 1(b). On average, the radiosondes reached an altitude of 23.9 km. For all soundings, both ascent and descent were fully recorded. One radiosonde balloon burst at an altitude of only 7 km, presumably due to lightning, and was excluded from the data set.

We used the German Weather Service's launch container on board the RV Maria S. Merian and the Vaisala DigiCORA MW 41 Sounding System, consisting of a workstation PC, the sounding processing system SPS311G, the ground check sub-system RI41 and a portable antenna set CG31. The workstation was connected to an automatic weather station next to the launch container for radiosonde initialization. Surface winds and sea surface temperature were obtained from DShip. The Vaisala radiosondes (RS41-SGP) were attached to helium-filled balloons. Most of them contained a parachute to slow down the descent (Totex TA200-No. 088). In some cases, however, the parachute did not open and the radiosonde fell freely.

The RS41-SGP radiosondes measure temperature (t_air) with an accuracy (combined sounding uncertainty) of $\pm 0.3$ K below 16 km and $\pm 0.4$ K above 16 km, pressure (p_air) with an accuracy of $\pm 1$ hPa above 100 hPa and $\pm 0.6$ hPa between 3 and 100 hPa, relative humidity (rh) with an accuracy of 4%, wind speed and direction with an accuracy of $\pm 0.15$ m s$^{-1}$ and 2° respectively. The position is determined by GPS with selective availability disabled and positional dilution of precision (PDOP) less than 4. Data are transmitted to the sounding station on board at a frequency of 1 Hz. Dew point temperature (dp) and mixing ratio (mr) are derived from the measured values.

The post-processing of the radiosonde data was done using the Python package pysonde (Schulz, 2023). In addition to the original mwx files (Köhler et al., 2024b) we provide two datasets. The Level 1 data set (Köhler et al., 2024c) contains the radio soundings converted and combined into a netCDF file, with the coordinate level corresponding approximately to one-second data. Each sounding is separated into ascent and descent, as indicated by its identifier (e.g. RS001_up versus RS001_down). In addition to the above quantities, the dataset contains the rate of ascent or descent (dz) and the height above the reference ellipsoid (alt_WGS84). The Level 2 data set (Köhler et al., 2024d) is interpolated to height levels in 10 m increments. In addition to the Level 1 quantities, the horizontal wind components $u$ (u_air) and $v$ (v_air), the potential temperature (theta), and the specific humidity (q) are provided. Furthermore, the interpolation can be examined by the number of observations and the bin method of GPS (N_gps and m_gps) and PTU (N_ptu and m_ptu).

## 3.8 Uncrewed Aircraft Vehicle

To document the variablility of the lower troposphere and to test the feasibility to perform measurements of stratification across an air-sea interface unperturbed by the vessel, we used an uncrewed aircraft vehicle (UAV) system with four different payloads

to sample the lower atmospheric boundary layer and the upper ocean during CTD times whenever weather conditions allowed. The positions of the UAV stations are indicated by orange triangles in Figure 1(c).

The UAV platform used during ARC was a small multirotor (quadcopter) remotely piloted aerial system with take-off mass below 1 kg manufactured by DJI Mavic Pro. The atmospheric payload was attached on top of the quadcopter on a specially designed 3D printed platform to ensure its safety and leveling. Typically, measurement flights lasted 10–15 min. The operational range was limited to a height of 500 m from the starting point on the ship, i.e. 510 m above sea level (ASL). The atmospheric UAV data are published in Baranowski et al. (2024a) and the oceanic data in Baranowski et al. (2024b).

### UAV payloads

There were three atmospheric packages (1-3) and one ocean package (4) of research equipment for the UAV:

1. the i-Met XQ2 package for profiling temperature (t_air) with accuracy $\pm$0.3 K, humidity (rh, q) with accuracy $\pm$5% and pressure (p_air) with accuracy $\pm$1.5 hPa between 5 m and 520 m ASL (1 Hz measurements);

2. the Trisonica Mini package for profiling temperature (t_air_trisonica), humidity (rh_trisonica, q_trisonica), pressure (p_air_trisonica) and horizontal wind speed (wspd, wdir) with accuracies in the range of 0.2–4% between 5 m and 510 m ASL (10 Hz measurements), extended by slower (1 Hz) measurements of temperature (t_skh), humidity (rh_skh, q_skh) and pressure (p_skh) from the SKH1 sensor with accuracies of $\pm$0.2 K, $\pm$1.8% and $\pm$1 hPa, respectively;

3. the OPC-N3 particle monitor (OPC) package for measuring the size of aerosol particles in the range of 0.35–40 $\mu$m (pm1, pm2d5, pm10) between 5 m and 410 m ASL (1 Hz measurements) extended by temperature, humidity and pressure measurements from SKH1 sensor, same as in the Trisonica Mini Package;

4. the RBR package for profiling oceanic temperatures from the surface down to 15 m with a RBR Solo3T sensor (operating at 2 Hz temporal resolution with the accuracy of $\pm$0.02 K) tethered on a 20 m long line underneath the UAV.

The atmospheric payloads (1-3) were placed on top of the UAV to avoid contamination from its dark frame and the radiators located in its lower part. Thus measurements during ascents are collected in an undisturbed environment while measurements during descents may be contaminated by the diabatic heating from the UAV frame. Hence, only ascent data are used in the atmospheric profiling.

### UAV operations and data processing

During ARC, all UAV measurements were performed during CTD stations, which frequently also aligned with radiosonde launches and allowed to obtain simultaneous data from both atmospheric and oceanic environments. The UAVs took off and landed at the helipad (approx. 10 m above sea level) near the bow of the vessel and measurements were collected in a distance of 50–100 m from the ship in an up-wind direction to ensure undisturbed conditions. During measurements, the UAV was positioned, so that sensors were also upwind relative to the drone's body. All UAV operations were performed when the sustained wind near the surface was below 10 m s$^{-1}$ and there was no precipitation. Different measurement payloads were

used during different flights, but typically 3–4 consecutive flights with different payloads were performed during a single CTD station. During ARC, a total of 120 flights (12 oceanic profiles and 108 atmospheric profiles) at 40 CTD stations were success-
340 fully performed. Atmospheric profiles include 70 temperature/humidity profiles with imet-XQ2 package, 24 combined profiles with SHK1 (temperature/humidity) and OPC-N3 (aerosols), and 14 combined profiles with SKH1 (temperature/humidity) and Trisonica Mini (horizontal winds). The oceanic profiles are published in a separate data set.

Data post-processing included bias corrections for pressure, temperature and humidity sensors, translation of horizontal wind data from measurements in UAV frame of reference to planetary frame of reference and binning the data in vertical
(above sea level altitude) direction. It should be noted that post-processed profiles of temperature, humidity and winds from all sensors are based on ascents only, while post-processed wind and aerosol data are based on measurements from hoovers only. All available data from a single flight were used to calculate a single, post-processed profile. The detailed description of flight types, measurement stratedy and data post processing is provided in appendix B

## 3.9 CTD

During ARC, a total of 43 CTDs were conducted, 41 of them to depths of 500 m, one to 3592 m on 31 January at 12:40 and one to 3791 m on 13 February at 16:00 (Lobo et al., 2024). For most of the days prior to 18 February, the vessel stopped twice a day to deploy CTDs. The positions of the CTDs are indicated by yellow triangles in figure 1(c). The SBE 911plus, SN-0807 CTD system has built-in sensors to measure pressure (p_sw), temperature (t_sw), conductivity and thereby salinity, oxygen saturation (oxygen), nitrogen saturation (nitrogen), fluorescence and turbidity. The dataset also contains the seawater density
(rho_sw) and the position (lat, lon) from DShip. Pressure is measured using a Digiquartz sensor. Temperature is measured with two SBE 3 sensors with an initial accuracy of $\pm0.001$ K, conductivity with an initial accuracy of $\pm0.0003$ S m$^{-1}$ with two SBE 4. Oxygen concentration is measured with two SBE 43 with an initial accuracy of $\pm2\%$ saturation. Fluorescence and turbidity are measured with a WETLabs ECO FLNTUR(TR)D sensor. Note that cholorophyll is not included in the data set. The whole system is mounted on an iron rosette with 24 Niskin bottles (10 l each) for collecting water samples. A QSP2300
Photosynthetically Active Radiation (PAR) sensor, designed to measure light available for photosynthesis from all directions, was mounted on top of the rosette. It was used for profiles to a depth of 2000 m.

For all measurements, the CTD was first lowered to 10 m below the surface by the winch operator to start the pump and remove any air bubbles from the bottles and sensors. After a few seconds, all sensors had stabilised under water and unwanted peaks in the measurement parameters were minimized. The CTD was then pulled up by the winch operator to just below the
365 surface and later lowered at a vertical speed of 0.3 - 1 m s$^{-1}$. After reaching the maximum depth, the CTD was pulled back to the surface, stopping at pre-determined depths to collect water samples. The data were interpolated to 1 m depth intervals.

## 4 Data collection

The main objective of this study is to make the data collected by the instruments listed above as useful as possible for future research. To this end, the data is published in a FAIR way, i.e., Findable, Accessible, Interoperable, and Re-usable (Wilkinson

et al., 2016). We distinguish between two different types of data sets. The first data type is data from instruments that measure continuously in time, i.e. DShip observations, HATPRO, and Ceilometer, as well as DustTrak. For the latter, it should be noted that there are gaps in the data at night and when it rained as the instrument was not waterproof. In order to make best use of the variety of instruments used during ARC, special attention has also been given to presenting data from different instruments in a way that facilitates comparison between them. To this end, the continuous data are averaged to a frequency of 1 minute

as a compromise between the different temporal resolutions of the different instruments. The second data type are the non-continuous data, which we call point measurements. Point measurements come from UAVs, CTDs, radiosondes, Calitoo, and Microtops. For instruments that have a launching or starting time, namely radiosondes, UAVs, and CTDs, we call this time stamp "start_time". Radiosondes also record the flight time which we simply call "time". For the point measurements, we provide (at least) one data set per instrument but, as mentioned above, we provide two data sets for the radio soundings. Level

1 data gives the single flights as function of time, in the level 2 data they are interpolated to 10 m altitude steps using pysonde (Schulz, 2023). Additionally, we provide the original level 0 mwx files for the radiosondes.

Table 2 gives an overview of the measured quantities. An extended version with all measured variables including standard deviations is given in the Appendix, see Table A1. The "name in data set" is the short name of the variable used in the netCDF file. Where available, the standard deviation is abbreviated to std. The quantity is marked as a "surface" variable if it has

no vertical resolution, i.e. it is a scalar number for a given location and time of the ship. "Profile" means that the variable is available as a function of height or depth. Some variables are measured by several instruments, some of them provide vertically resolved data. For example, relative humidity is provided as a surface variable by DShip and as a profile variable by HATPRO, UAVs, and radiosondes.

All data can be downloaded from Pangaea (Köhler et al., 2024a). The netCDF files contain the post-processed, quality

checked data. We have removed unphysical values as mentioned in the instrument descriptions in Section 3. Where necessary, we have interpolated the data to regular altitude levels to simplify working with the data sets. Furthermore, as mentioned above, we have standardised the variable names for the different instruments according to Table 2. The UAVs measure some of the meteorological variables with different sensors, all of which are listed in the same file. In this case we have added the instrument name to the short name for all sensors except the one with maximum coverage. The post-processing scripts

and some minimal examples can be found in the corresponding GitHub repository (Köhler, 2023). It contains the settings used for shipspy and pysonde, the Python environment, the renaming dictionaries including variable and global attributes, a reprocessing script and some basic information about the campaign. Each variable is given a short name, see second column in table 2, a long descriptive name and, if available, the standard name according to the standard name table (Climate and Forecast (CF) Conventions Committee and Standard Names Committee, 2023) of the NetCDF Climate and Forecast (CF)

Metadata Conventions (Eaton et al., 2022). If the CF standard name table does not contain the variable, the standard name is omitted. We use SI units according to the CF standard name table. In addition, we indicate in the attributes the instrument with which the quantity was observed and, if applicable, comments on the processing, etc.

The continuous data are provided for the period when the research vessel was in international waters, i.e. from 7am on 25 January to 3pm on 20 February. HATPRO data are only included up to 15 February. Point data are available for the times of the

measurements. Radiosondes have already been launched in the national waters of Cape Verde thanks to a special measurement permit. All data sets have an auxiliary coordinate called "section" to facilitate the handling of the data, in particular to study the different ITCZ crossings. The section coordinate can be added using shipspy (Köhler, 2024), for which an example file specifying the sections of ARC can be found in the GitHub repository (Köhler, 2023). For ARC, we have split the data into five sections. Section 0 contains everything before we entered the ITCZ, sections 1 to 3 correspond to the three crossings, and section 4 contains the data from the transit period after the ITCZ crossings.

As stated in the introduction, an additional goal to publishing the data collected during ARC is to provide a blueprint for publishing ship campaign data in a standardised and user-friendly way. To this end, the approach presented above, based on the Python package shipspy and the GitHub repository, can be adopted for future ship campaigns. In particular, the GitHub repository can be cloned and adapted to the specific instrumentation on board.

Table 2: Overview of variables and their short names used in the NetCDF data sets. Surface corresponds to 1d (scalar) variables, whereas profile marks variables which depend on height or depth. The complete version is given in Table A1 where also standard deviations and additional parameters are included.

| Variable | Name in data set | surface | profile | DShip | HATPRO | Ceilometer | DustTrak | Radiosondes | UAV | CTD | Calitoo | Microtops |
|---|---|---|---|---|---|---|---|---|---|---|---|---|
| **Time and position** | | | | | | | | | | | | |
| time | time | ● | ● | ● | ● | ● | ● | ● | | | ● | ● |
| | start_time | ● | | | | | | ● | ● | ● | | |
| latitude | lat | ● | ● | ● | ● | ● | ● | ● | ● | ● | ● | ● |
| longitude | lon | ● | ● | ● | ● | ● | ● | ● | ● | ● | ● | ● |
| altitude | alt | | ● | | | ● | | ● | ● | ● | | |
| depth | depth | | ● | | | | | | | ● | | |
| campaign section | section | ● | ● | ● | ● | ● | ● | ● | ● | ● | ● | ● |
| **Meteorology** | | | | | | | | | | | | |
| air temperature | t_air | ● | ● | ● | ● | | | ● | ● | | ● | |
| air temperature std | t_air_std | | ● | | | | | | ● | | | |
| potential temperature | theta | | ● | | | ● | | ● | | | | |
| dew point temperature | dp | | ● | | | | | ● | | | | |
| air pressure | p_air | ● | ● | ● | | | | ● | ● | | ● | |
| wind speed | wspd | ● | ● | ● | | | | ● | ● | | | |
| wind direction | wdir | ● | ● | ● | | | | ● | ● | | | |
| relative humidity | rh | ● | ● | ● | ● | | | ● | ● | | | |

| Variable | Name in data set | surface | profile | DShip | HATPRO | Ceilometer | DustTrak | Radiosondes | UAV | CTD | Calitoo | Microtops |
|---|---|---|---|---|---|---|---|---|---|---|---|---|
| absolute humidity | ah | | ● | | ● | | | | | | | |
| specific humidity | q | | ● | | | | | ● | ● | | | |
| mixing ratio | mr | | ● | | | | | ● | | | | |
| column-integrated water vapor | cwv | ● | | | ● | | | | | | | ● |
| liquid water path | lwp | ● | | | ● | | | | | | | |
| long wave radiation | lwr | ● | | ● | | | | | | | | |
| short wave radiation | swr | ● | | ● | | | | | | | | |

**Aerosols**

| Variable | Name in data set | surface | profile | DShip | HATPRO | Ceilometer | DustTrak | Radiosondes | UAV | CTD | Calitoo | Microtops |
|---|---|---|---|---|---|---|---|---|---|---|---|---|
| PM1 ambient aerosols | pm1 | ● | ● | | | | ● | | ● | | | |
| PM2.5 ambient aerosols | pm2d5 | ● | ● | | | | ● | | ● | | | |
| PM4 ambient aerosols | pm4 | ● | | | | | ● | | | | | |
| PM10 ambient aerosols | pm10 | ● | ● | | | | ● | | ● | | | |
| Total ambient aerosols | pm_all | ● | | | | | ● | | | | | |
| aerosol optical thickness 380 nm | aot_380 | ● | | | | | | | | | | ● |
| aerosol optical thickness 440 nm | aot_440 | ● | | | | | | | | | | ● |
| aerosol optical thickness 465 nm | aot_465 | ● | | | | | | | | | ● | |
| aerosol optical thickness 500 nm | aot_500 | ● | | | | | | | | | | ● |
| aerosol optical thickness 540 nm | aot_540 | ● | | | | | | | | | ● | |
| aerosol optical thickness 619 nm | aot_619 | ● | | | | | | | | | ● | |
| aerosol optical thickness 675 nm | aot_675 | ● | | | | | | | | | | ● |
| aerosol optical thickness 870 nm | aot_870 | ● | | | | | | | | | | ● |
| air mass | air_mass | ● | | | | | | | | | | ● |
| assimilated total ozone | ato | ● | | | | | | | | | ● | |
| Angstrom exponent | angstrom_exp | ● | | | | | | | | | ● | ● |
| aerosol layer in PBL | pbl | | ● | | | ● | | | | | | |

**Clouds**

| Variable | Name in data set | surface | profile | DShip | HATPRO | Ceilometer | DustTrak | Radiosondes | UAV | CTD | Calitoo | Microtops |
|---|---|---|---|---|---|---|---|---|---|---|---|---|
| cloud cover (total) | tcc | ● | | | | ● | | | | | | |
| cloud cover (base) | bcc | ● | | | | ● | | | | | | |
| cloud base height | cbh | ● | | | | ● | | | | | | |
| cloud base height (time dependent) | cbh_2s | ● | | | | ● | | | | | | |
| cloud depth | cdp | ● | | | | ● | | | | | | |

| Variable | Name in data set | surface | profile | DShip | HATPRO | Ceilometer | DustTrak | Radiosondes | UAV | CTD | Calitoo | Microtops |
|---|---|:---:|:---:|:---:|:---:|:---:|:---:|:---:|:---:|:---:|:---:|:---:|
| sky condition index | sci | ● | | | | ● | | | | | | |
| vertical optical range | vor | | ● | | | ● | | | | | | |
| **Sea water** | | | | | | | | | | | | |
| sea surface temperature | sst_2m | ● | | ● | | | | | | | | |
| | sst_7m | ● | | ● | | | | | | | | |
| sea water temperature | t_sw | | ● | | | | | | | ● | | |
| sea water pressure | p_sw | | ● | | | | | | | ● | | |
| sea water density | rho_sw | | ● | | | | | | | ● | | |
| sea floor depth | sea_floor_depth | ● | | ● | | | | | | | | |
| conductivity | conductivity | ● | ● | ● | | | | | | ● | | |
| salinity | salinity | ● | ● | ● | | | | | | ● | | |
| sea water turbidity | turbidity | | ● | | | | | | | ● | | |
| oxygen saturation | oxygen | | ● | | | | | | | ● | | |
| nitrogen saturation | nitrogen | | ● | | | | | | | ● | | |
| fluorescence | fluorescence | | ● | | | | | | | ● | | |
| chlorophyll A | chlorophyll_a | ● | | ● | | | | | | | | |
| sea water speed | curernt_speed | ● | | ● | | | | | | | | |
| sea water direction | current_dir | ● | | ● | | | | | | | | |
| **Waves** | | | | | | | | | | | | |
| wave mean period | wave_period | ● | | ● | | | | | | | | |
| wave maximum wavelength | wave_length | ● | | ● | | | | | | | | |
| wave significant height | wave_height | ● | | ● | | | | | | | | |
| wave direction | wave_dir | ● | | ● | | | | | | | | |
| **Ship orientation** | | | | | | | | | | | | |
| ship speed | ship_speed | ● | | ● | | | | | | | | |
| ship heading | ship_heading | ● | | ● | | | | | | | | |
| ship heave | ship_heave | ● | | ● | | | | | | | | |
| ship pitch | ship_pitch | ● | | ● | | | | | | | | |
| ship roll | ship_roll | ● | | ● | | | | | | | | |

## 5 Measurement examples

In this section we give an overview of the observations collected during ARC and discuss them at a phenomenological level. We focus on the three ITCZ crossings, but also show data from the storm in the Roaring Forties. During the storm, extreme weather conditions prevented deck measurements such as CTDs or radiosondes, so we were limited to data from the ship's integrated instruments for this period. We also present vertical profiles for the entire duration of the campaign.

### 5.1 Crossings of the ITCZ

The Atlantic ITCZ exhibits a rich dynamical and thermodynamic structure (Windmiller and Stevens, 2024), which we also observed during our crossings. Following Mapes et al. (2018) and Windmiller and Stevens (2024), we will use the nomenclature below to define the edges of the moist tropics as well as the edges of the ITCZ within the moist tropics. According to Mapes et al. (2018), we define the edges of the moist tropics as the northernmost and southernmost latitudes where the water vapour column is 48 mm. Within the moist tropics, we define the northern and southern edges of the ITCZ based on surface convergence. Following Windmiller and Stevens (2024), we refer to the confluence line as the northern edge of the ITCZ and the speed convergence line as the southern edge of the ITCZ. The confluence line is the northernmost latitude at which the meridional wind speed changes sign. The speed convergence line is the latitude at which the meridional wind speed decreases most rapidly. During our crossings, the edges of the ITCZ were often accompanied by deep, precipitating convection. The edges of the ITCZ described above are easily visible in Fig. 2 as they are marked by two lines of convection extending from Africa to South America. The northern edge, north of the ship's position, is particularly pronounced, with several deep convective clusters in a row. The southern edge, which the RV Maria S. Merian had just entered, has less deep convection but can still be seen as a band of clouds in the satellite image. Note that the ITCZ and its structure is not always as clear as in this example. Within the ITCZ, i.e. between the edges, we found regions with low and variable wind speeds, often with few and shallow clouds and somewhat reduced CWV. We refer to these regions as doldrums (e.g. Klocke et al., 2017; Windmiller, 2024).

Figure 4 (a)-(c) shows data from the three ITCZ crossings as a function of time. The respective latitudes are marked on the top x-axis. The upper panel always shows the air temperature from DShip in dark blue and the column integrated water vapour (CWV) from the HATPRO in light blue. The blue stars correspond to the CWV derived from radiosonde ascents by integrating the water content. To make sure that the integration starts at sea level, we restrict ourselves to ascents. The blue crosses show the column-integrated water vapour from the Microtops observations. The dashed grey line at 48 mm serves as a guide. The light grey bars mark regions where precipitation occurred according to the HATPRO. The middle panel shows the horizontal wind components derived from the wind speed and direction after taking the 1 minute mean of the DShip data. Light red shows the zonal wind $u$ and dark red shows the meridional wind $v$. The bottom panel is different for each crossing.

We define the start of the first crossing, marked as section 1, as 1am on 26 January and the end of the first crossing as 3am on 30 January. During this time we went from 9°N to 5°S at an average longitude of 22.9°W. We passed the PIRATA buoy at 4°N 23°W. Figure 4(a) shows data from the first crossing. As shown in light blue in the top panel, the CWV increased very slowly as we entered the moist tropics from the north. We first encountered deep precipitating convection about 4° south

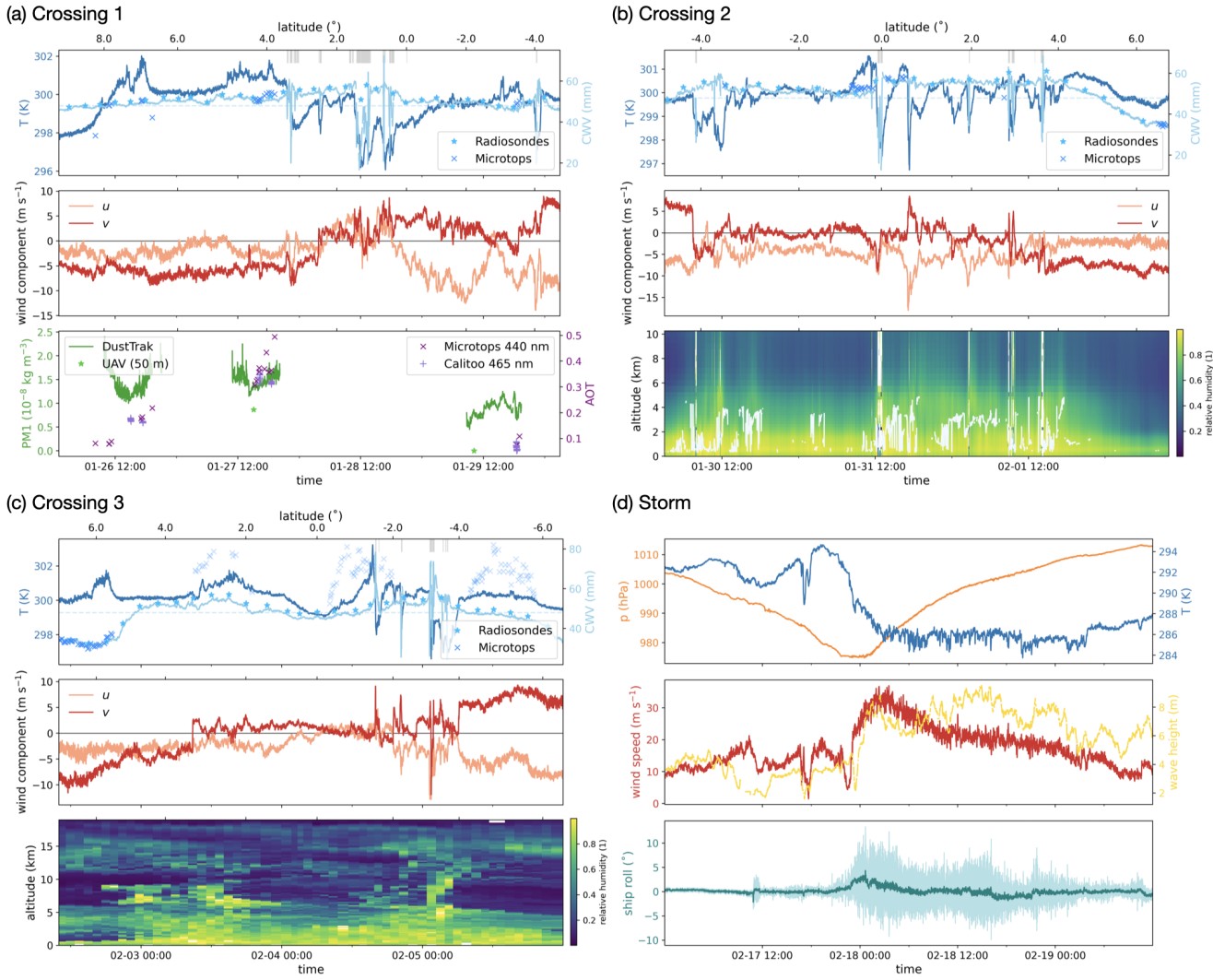

**Figure 4.** (a)-(c) ITCZ crossings. The upper panel shows temperature (dark blue) and column-integrated water vapor (light blue line from HATPRO, blue stars from radiosonde ascents, blue crosses from Microtops). Regions with precipitation are marked by gray bars. The second panel shows the horizontal wind components $u$ (light red) and $v$ (dark red). (a) The third panel presents aerosol data. Green shows the particle mass concentration with diameters below 1 micron. The green line is data from DustTrak and the light green crosses are measured with UAVs at 50 m height. The purple crosses represent measurements of the aerosol optical depth at 440 nm from the Microtops (dark purple) and at 465 nm from the Calitoo (light purple), respectively. (b) The third panel shows relative humidity profiles from the HATPRO. The white curve presents the cloud base height from the Ceilometer. (c) The third panel shows the relative humidity from the radiosonde ascents and descents. (d) DShip observations during the storm. The first panel shows pressure (orange) and air temperature (blue). The second panel presents the wind speed (red) and the wave height (yellow). The lower panel displays the ship roll. The lighter shaded area represents the standard deviation and the darker line shows the mean.

of the moist edge, crossing the northern edge of the ITCZ. Inside the ITCZ, the wind speed decreased, the wind direction changed, and between 1°N and 2°S we crossed a region of particularly low and variable wind speeds. During this time, we crossed precipitating convection and observed lightning. The southern edge of the ITCZ moved ahead of us and developed into a large convective feature. We did not leave the ITCZ until we had also crossed the southern edge of the moist tropics south of 4°S and entered the trade wind region just before turning back. The bottom panel shows aerosol measurements taken during the first crossing. The DustTrak continuously samples the aerosol concentration of the ambient air. These surface data are often decoupled from the total column AOT values measured by the sunphotometers. Sunphotometer observations require strict cloud filtering; the period 26-29 January was quite cloudy, so there are few data points. On 26 January a desert dust plume created elevated AOT values at 465 nm (Calitoo) and 440 nm (Microtops) of about 0.17, while on 29 January background AOT values of about 0.06 occurred.

Figure 4(b) shows the second crossing. It took place from 3am on 30 January to 10am on 2 February, going from 5°S to 6.5°N at 22.85°W. Shortly after the start of the second crossing we crossed the southern edge of the moist tropics and the southern edge of the ITCZ. At about 4°S we entered a region of very low wind speeds. The deep convection associated with the southern edge of the ITCZ during the first crossing had dissipated. On our way north, a new southern edge formed in front of us and we crossed precipitating convection close to the equator. There was then another region of particularly low meridional wind speeds before entering another convective region between 3°N and 4°N, associated with the northern edge. This was followed by the entry into the northerly trade winds with constant wind speeds. The CWV also dropped below 48 mm. The bottom panel shows the relative humidity from the HATPRO profiles. Particularly high humidity values up to altitudes of 10 km occurred at the edges of the ITCZ. In the low wind speed regions, the relative humidity is lower than at the edges, especially above 5 km. The northern trade wind region is very dry with very little humidity above 2 km. Note that the HATPRO profiles are more accurate below 4 km than above as discussed in Section 3.2. The light grey line shows the cloud base height of the lowest layer from the ceilometer. In the convective regions, the cloud base height was between 322 m and 10 km. In the northerly trade winds, the sky was clear, and in the low wind speed region south of the equator, there were only a few shallow clouds.

The third crossing, shown in figure 4(c), took place from 2 February 10 am to 6 February 12 am and spanned the latitudes from 6.5°N to 6.5°S at 23.16°W. Coming from the north, we observed a steep increase in CWV as we entered the moist tropics at 5°N. The wind direction changes at about 3°N. In this region we find an enhanced relative humidity in the radiosonde profiles shown in the bottom panel up to an altitude of 10 km. During this crossing we observed very low wind speeds between 3°N and 1.5°S and almost no convection. Interestingly, we find that the CWV again decreased in this region, even falling below 48 mm. The decrease in CWV observed during crossing 3 coincides with a lower relative humidity in the radiosonde profiles. At 1.5°S we entered the southern edge of the ITCZ with deep precipitating convection. For a while, the edge moved with us at about the same speed as the ship, and we remained in a region of deep convection until 4°S. Then we entered the southerly trade winds and the CWV dropped below 48 mm. During this third crossing, we observed a peak in aerosols with values of the aerosol optical thickness measured with the Microtops at 380 and 440 nm exceeding 0.7 due to smoke coming from Africa (not shown).

Comparing the CWV from the HATPRO with the radiosondes during the crossings, we find that both instruments are in good agreement with a root mean square deviation (RMSD) of 4.3 mm. The CWV from the radiosondes, however, tends to be slightly higher which is likely due to the fact that it includes the whole troposphere whereas the HATPRO only integrates up to 10 km height. Furthermore, the radiosondes are not affected as much as the HATPRO by precipitation events and thus do not show excessively large signals during rain. The Microtops uses the 940 nm absorption line to determine the column water vapour in clear sky. In crossing 1 and 2 and at the beginning of crossing 3, it coincides closely with the other two instruments. The RMSD of the Microtops CWV from the HATPRO CWV until 2 February is 2.7 mm. However, in the course of crossing 3 from 3 February on, the Microtops shows very high CWV values and a RMSD from the HATPRO of 19.2 mm. This, together with the precipitation rate as indicated by the HATPRO and the cloud base height values detected with the ceilometer, suggests that the conditions were not completely cloud free despite the post processing and shows the high sensitivity of the Microtops to the presence of clouds. We therefore show the CWV values from the Microtops after 3 February only with light blue crosses. In summary, we find that the HATPRO CWV coincides very well especially with the radiosoundings but also most of the time with the Microtops. This comparison illustrates more generally how the availability of data in a consistent way not only facilitates the comparison of different quantities, but also the comparison of measurements of the same quantity made by different instruments to assess the quality of the data.

This initial analysis of the data shows that during all three crossings we passed through regions of very low surface wind speeds, typical of the doldrums, surrounded by regions of deep convection and variable wind. Most of the precipitation occurred outside the regions of low wind speeds at the edges of the ITCZ, indicated by strong changes in the meridional wind speed as described in Windmiller and Stevens (2024).

### 5.2 Storm

Figure 4(d) shows DShip data from the storm we crossed in the Roaring Forties between 12am on 17 February and 12pm on 19 February. During this time we moved from 45°W 40°S to 54°W 45°S. The centre of the low-pressure system was south-east of us and moved further east on 19 February. The orange line in the upper panel of figure 4(d) shows the on-board surface pressure measurements, which reached a minimum of 974.8 hPa at 00:30 on 18 February. This was accompanied by a sudden temperature drop of 10 K, shown by the blue line. The middle panel shows the wind speed in red. The maximum 1-minute average wind speed was 39 m s$^{-1}$. The yellow line in the middle panel shows the significant wave height determined by WaMoS with a maximum of 9.5 m. Under these conditions there was naturally a large increase in the heave, pitch and roll of the ship. The bottom panel shows the 1 minute averaged values of the roll in turquoise with a maximum of 4.3°. The lighter shaded area around the curve marks the standard deviation from the mean for each minute. It reached a maximum of $\pm 10.8°$. After the storm, the temperature remained low and the wind speed exceeded 10 m s$^{-1}$ until the morning of 20 February.

### 5.3 Profiles

Figure 5 shows relative humidity profiles from UAVs, HATPRO and radiosondes and sea water temperature profiles from CTDs for the entire period of the campaign. As UAV flights were only possible when conditions allowed, there are at most two

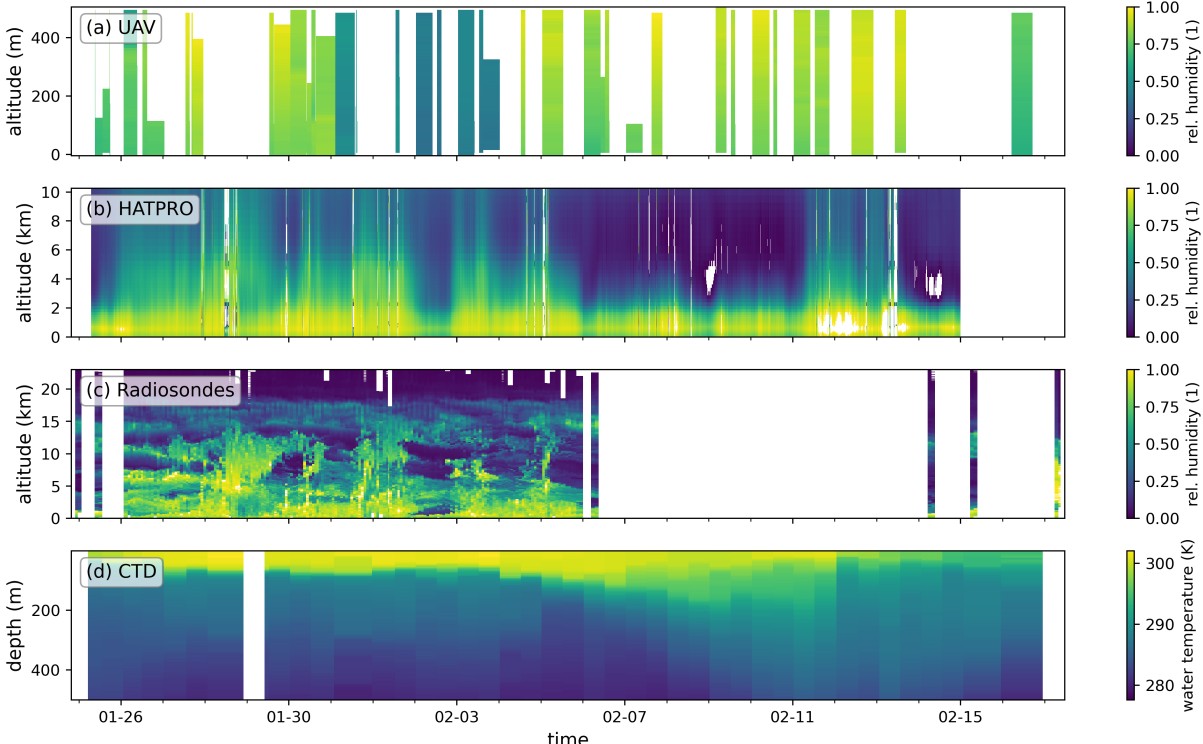

**Figure 5.** Relative humidity profiles during ARC (a, b, c) and sea water temperature profiles (d). The relative humidity is measured by (a) UAVs (using the i-Met XQ2 package) up to a height of 500 m, by (b) the HATPRO up to 10 km, and (c) by radiosondes up to around 23 km. (d) shows CTD profiles of sea water temperature up to a depth of 500 m.

profiles per day, but also longer intervals between flights, see Fig. 5(a). The UAV profiles reach up to 500 m, i.e. they scan the lower boundary layer. Figure 5(b) shows the HATPRO relative humidity up to 10 km. The UAVs only measure the lowest part of the HATPRO profiles. The HATPRO measured continuously. There are a few data gaps due to non-physical values (white regions) until the data stops on 15 February as data quality could not be guaranteed for later times. The radiosondes provide
profiles of relative humidity up to about 23 km, but were only launched regularly within the moist tropics, see figure 5(c). Comparing the wind speeds plotted in Figure 4 with the relative humidity shown in Figure 5, we find that, as noted above and illustrated in Figure 4(c), low wind speed events tended to coincide with reduced humidity, especially in the free troposphere.

Before and after the ITCZ crossings, only a few radiosondes were launched coordinated with transits of the AEOLUS satellite to sample reference data. In the HATPRO data, we can see that the atmosphere gets drier outside the moist tropics although the
UAV data shows that the lower boundary layer remained moist until 14 February. In the region of 11 to 12 February, we passed a more humid region, also shown in the HATPRO data available until 15 February.

Due to the decreasing accuracy of the HATPRO with height (see section 3.2), the quantitative HATPRO profiles are more reliable below four km than above. This is also reflected when comparing the HATPRO and radiosonde relative humidity by calculating the RMSD. Below four km, we find it to be 0.13 whereas in the range between 4 and 10 km it is 0.22.

The bottom panel (d) shows CTD profiles of the ocean down to 500 m which is the maximum depth of the CTD scans with two exceptions as mentioned above. On most days, two CTDs were deployed. The plot shows the sea water temperature. We find much more mixing after 7 February, corresponding to a latitude of 10°S. The sea surface temperature outside the tropics decreases as expected from 301.9 K on 3 February at 2.3°N to 292.4 on 17 February at 41°S. Due to better mixing in the extra-tropics, the maximum temperature at 500 m depth of 284.6 K occurs on 11 February at 26.6°S, while in the tropics we
find a minimum temperature at 500 m depth of 280.0 K on 5 February at 5.3°S.

## 6  Conclusions

This article presents the atmospheric and oceanographic observations from the Atlantic References and Convection (ARC) campaign, with reference MSM114/2, with the German ice margin research vessel Maria S. Merian. During the campaign, the research vessel crossed from Mindelo, Cape Verde, to Punta Arenas, Chile. One major focus of ARC was on the Atlantic
Intertropical Convergence Zone (ITCZ) which was crossed three times in meridional direction with the northern and southern turning points adjusted to the latitudinal position of the ITCZ. The longitude of the cross sections (23°W) was chosen such that the collected data can be linked to long-standing buoy measurements through the PIRATA array. Towards the end of the campaign, we were forced to deviate from our original track as we crossed a storm in the South Atlantic. Even so, the pressure minimum we passed was 975 hPa and the maximum 1-minute averaged wind speed 39 m s$^{-1}$.

A set of continously operating instruments provided (vertically resolved) cross sections of the atmosphere and upper ocean from 25 January 7am to 20 February 3pm while the ship was in international waters. Key instruments included a microwave radiometer (cloud water and atmospheric water content), a ceilometer (cloud base height and areosol profiles), an aerosol monitor (aerosol mass measurements), and sunphotometers (aerosol and water vapour column properties). These measurements were supported by the complementary operational instruments on the ship (e.g. near-surface air and water temperature, relative
humidity, wind direction and speed). For additional vertical profiling we used CTD (Conductivity, Temperature and Depth) measurements (mainly of the upper 500 m of the ocean), uncrewed aircraft vehicles (humidity, temperature, aerosols and wind up to 500 m), and radiosondes (humidity, temperature and wind up to 25 km) with the radiosonde launches largely limited to three hourly launches within the ITCZ.

A key motivation for this study was to present the collected data in a consistent, easy to use and accessible manner. The data
can be downloaded from Köhler et al. (2024a) as NetCDF files. They provide an opportunity to study the (tropical) Atlantic with standardised datasets, facilitating the comparison and combination of different instruments. We hope that this approach might serve as a blueprint for future (and past) ship campaigns. To this end, this article is complemented by the Python package shipspy (Köhler, 2024) and a GitHub repository (Köhler, 2023) which provides detailed information about the settings for

shipspy and pysonde (Schulz, 2023) and additional processing scripts. This repository could be cloned and adjusted for future campaigns.

As a first overview of the collected data, we present measurements of the three crossings of the ITCZ as well as of the South Atlantic storm. As expected from previous studies, the ITCZ is characterised by increased column water vapour and its edges by steep gradients in meridional wind speed. Interestingly, we found regions of very low wind speeds accompanied by a slight decrease in column water vapour between the ITCZ edges, which we associate with the doldrums. As the doldrums are poorly understood, these low wind speed events are investigated in Windmiller (2024). The doldrums are just one example that illustrates our lack of knowledge about the dynamic and thermodynamic properties of the Atlantic ITCZ. Future (ship) campaigns are planned to complement the measurements collected and presented here. The application of the presented standardised data processing approach will not only allow the comparison of measurements from different instruments within a campaign, but also facilitate the comparison of measurements from different campaigns.

*Code availability.* The code for the post processing of the raw data can be downloaded from the following GitHub repository: Köhler (2023). Standardisation of the different instruments was done with the python package shipspy (Köhler, 2024). The HATPRO data were processed with the MWRpy package (Marke et al., 2024). The radiosoundings were processed with the pysonde package (Schulz, 2023).

*Data availability.* The processed data can be downloaded from Köhler et al. (2024a) with doi https://doi.pangaea.de/10.1594/PANGAEA. 966616 where the data sets for the single instruments are collected and linked (Stammes et al., 2024a; Köhler et al., 2024e, f; Kinne and Köhler, 2024a; Lobo et al., 2024; Köhler and Windmiller, 2024; Stammes et al., 2024b; Hayo et al., 2024; Kinne and Köhler, 2024b; Köhler et al., 2024b, c, d; Baranowski et al., 2024a, b).

## Appendix A: Variable overview

Here, we show the complete table with all variables included in the published data sets. It is the extended version of Table 2 with additional (technical) parameters and standard deviations.

Table A1: Overview of variables and their short names used in the NetCDF data sets. Standard deviation is abbreviated as std. Surface corresponds to 1d (scalar) variables, whereas profile marks variables which depend on height or depth. Table 2 in the main text is a shortend version of this table.

| Variable | Name in data set | surface | profile | DShip | HATPRO | Ceilometer | DustTrak | Radiosondes | UAV | CTD | Calitoo | Microtops |
|---|---|---|---|---|---|---|---|---|---|---|---|---|
| Time and position | | | | | | | | | | | | |

| Variable | Name in data set | surface | profile | DShip | HATPRO | Ceilometer | DustTrak | Radiosondes | UAV | CTD | Calitoo | Microtops |
|---|---|---|---|---|---|---|---|---|---|---|---|---|
| time | time | ● | ● | ● | ● | ● | ● | ● | | | ● | ● |
| | start_time | ● | | | | | | ● | ● | ● | | |
| latitude | lat | ● | ● | ● | ● | ● | ● | ● | ● | ● | ● | ● |
| longitude | lon | ● | ● | ● | ● | ● | ● | ● | ● | ● | ● | ● |
| altitude | alt | | ● | | ● | | | ● | ● | ● | | |
| | alt_sensor | ● | | | ● | | | | | | ● | |
| altitude bounds | alt_bnds | | ● | | | | | ● | | | | |
| depth | depth | | ● | | | | | | | ● | | |
| distance | range | | ● | | | | ● | | | | | |
| | range_hr | | ● | | | | ● | | | | | |
| campaign section | section | ● | ● | ● | ● | ● | ● | ● | ● | ● | ● | ● |
| Meteorology | | | | | | | | | | | | |
| air temperature | t_air | ● | ● | ● | ● | | | ● | ● | | ● | |
| | t_air_skh | | ● | | | | | | ● | | | |
| | t_air_trisonica | | ● | | | | | | ● | | | |
| air temperature std | t_air_std | | ● | | | | | | ● | | | |
| | t_air_skh_std | | ● | | | | | | ● | | | |
| | t_air_trisonica_std | | ● | | | | | | ● | | | |
| potential temperature | theta | | ● | | | ● | | ● | | | | |
| equivalent potential temperature | theta_e | | ● | | | ● | | | | | | |
| dew point temperature | dp | | ● | | | | | ● | | | | |
| air pressure | p_air | ● | ● | ● | | | | ● | ● | | ● | |
| | p_air_skh | | ● | | | | | | ● | | | |
| | p_air_trisonica | | ● | | | | | | ● | | | |
| air pressure std | p_air_std | | ● | | | | | | ● | | | |
| | p_air_skh_std | | ● | | | | | | ● | | | |
| | p_air_trisonica_std | | ● | | | | | | ● | | | |
| wind speed | wspd | ● | ● | ● | | | | ● | ● | | | |
| wind speed std | wspd_std | | ● | | | | | | ● | | | |
| wind direction | wdir | ● | ● | ● | | | | ● | ● | | | |
| zonal wind component | u_air | | ● | | | | | ● | | | | |

| Variable | Name in data set | surface | profile | DShip | HATPRO | Ceilometer | DustTrak | Radiosondes | UAV | CTD | Calitoo | Microtops |
|---|---|---|---|---|---|---|---|---|---|---|---|---|
| meridional wind component | v_air | | • | | | | | • | | | | |
| decent rate | dz | | • | | | | | • | | | | |
| relative humidity | rh | • | • | • | • | | | • | • | | | |
| | rh_skh | | • | | | | | | • | | | |
| | rh_trisonica | | • | | | | | | • | | | |
| relative humidity std | rh_std | | • | | | | | | • | | | |
| | rh_skh_std | | • | | | | | | • | | | |
| | rh_trisonica_std | | • | | | | | | • | | | |
| absolute humidity | ah | | • | | • | | | | | | | |
| specific humidity | q | | • | | | | | • | • | | | |
| | q_skh | | • | | | | | | • | | | |
| | q_trisonica | | • | | | | | | • | | | |
| specific humidity std | q_std | | • | | | | | | • | | | |
| | q_skh_std | | • | | | | | | • | | | |
| | q_trisonica_std | | • | | | | | | • | | | |
| mixing ratio | mr | | • | | | | | • | | | | |
| column-integrated water vapor | cwv | • | | | • | | | | | | | • |
| column-integrated water vapor std | cwv_std | • | | | | | | | | | | • |
| liquid water path | lwp | • | | | • | | | | | | | |
| liquid water path offset | lwp_offset | • | | | • | | | | | | | |
| long wave radiation | lwr | • | | • | | | | | | | | |
| short wave radiation | swr | • | | • | | | | | | | | |
| Aerosols | | | | | | | | | | | | |
| PM1 ambient aerosols | pm1 | • | • | | | | • | | • | | | |
| PM2.5 ambient aerosols | pm2d5 | • | • | | | | • | | • | | | |
| PM4 ambient aerosols | pm4 | • | | | | | • | | | | | |
| PM10 ambient aerosols | pm10 | • | • | | | | • | | • | | | |
| Total ambient aerosols | pm_all | • | | | | | • | | | | | |
| PM1 ambient aerosols std | pm1_std | | • | | | | | | • | | | |
| PM2.5 ambient aerosols std | pm2d5_std | | • | | | | | | • | | | |
| PM10 ambient aerosols std | pm10_std | | • | | | | | | • | | | |

| Variable | Name in data set | surface | profile | DShip | HATPRO | Ceilometer | DustTrak | Radiosondes | UAV | CTD | Calitoo | Microtops |
|---|---|---|---|---|---|---|---|---|---|---|---|---|
| aerosol optical thickness 380 nm | aot_380 | • | | | | | | | | | | • |
| aerosol optical thickness 440 nm | aot_440 | • | | | | | | | | | | • |
| aerosol optical thickness 465 nm | aot_465 | • | | | | | | | | | • | |
| aerosol optical thickness 500 nm | aot_500 | • | | | | | | | | | | • |
| aerosol optical thickness 540 nm | aot_540 | • | | | | | | | | | • | |
| aerosol optical thickness 619 nm | aot_619 | • | | | | | | | | | • | |
| aerosol optical thickness 675 nm | aot_675 | • | | | | | | | | | | • |
| aerosol optical thickness 870 nm | aot_870 | • | | | | | | | | | | • |
| aerosol optical thickness 380 nm std | aot_380_std | • | | | | | | | | | | • |
| aerosol optical thickness 440 nm std | aot_440_std | • | | | | | | | | | | • |
| aerosol optical thickness 500 nm std | aot_500_std | • | | | | | | | | | | • |
| aerosol optical thickness 675 nm std | aot_675_std | • | | | | | | | | | | • |
| aerosol optical thickness 870 nm std | aot_870_std | • | | | | | | | | | | • |
| air mass | air_mass | • | | | | | | | | | | • |
| assimilated total ozone | ato | • | | | | | | | | | • | |
| Angstrom exponent | angstrom_exp | • | | | | | | | | | • | • |
| Angstrom exponent std | angstrom_exp_std | • | | | | | | | | | | • |
| elevation angle (sensor) | elevation | • | | | • | | | | | | • | |
| azimuth angle (sensor) | azimuth | • | | | • | | | | | | | |
| aerosol layer in PBL | pbl | | • | | | • | | | | | | |
| Clouds | | | | | | | | | | | | |
| cloud cover (total) | tcc | • | | | | • | | | | | | |
| cloud cover (base) | bcc | • | | | | • | | | | | | |
| cloud base height | cbh | • | | | | • | | | | | | |
| cloud base height (time dependent) | cbh_2s | • | | | | • | | | | | | |
| cloud base height (layer 1, 2, 3) | cbh_layer | | • | | | • | | | | | | |
| cloud base height error (layer) | cbh_layer_error | | • | | | • | | | | | | |
| cloud depth | cdp | • | | | | • | | | | | | |
| cloud depth error | cdp_error | • | | | | • | | | | | | |
| sky condition index | sci | • | | | | • | | | | | | |
| vertical optical range | vor | | • | | | • | | | | | | |

| Variable | Name in data set | surface | profile | DShip | HATPRO | Ceilometer | DustTrak | Radiosondes | UAV | CTD | Calitoo | Microtops |
|---|---|---|---|---|---|---|---|---|---|---|---|---|
| vertical optical range error | vor_error | | ● | | | ● | | | | | | |
| maximum detection height | mxd | | ● | | | ● | | | | | | |
| back-scattered signal | beta_raw | | ● | | | ● | | | | | | |
| | beta_raw_hr | | ● | | | ● | | | | | | |
| raw signal standard deviation | std | ● | | | | ● | | | | | | |
| raw signal baseline | base | ● | | | | ● | | | | | | |
| Sea water | | | | | | | | | | | | |
| sea surface temperature | sst_2m | ● | | ● | | | | | | | | |
| | sst_7m | ● | | ● | | | | | | | | |
| sea water temperature | t_sw | | ● | | | | | | | ● | | |
| sea water pressure | p_sw | | ● | | | | | | | ● | | |
| sea water density | rho_sw | | ● | | | | | | | ● | | |
| sea floor depth | sea_floor_depth | ● | | ● | | | | | | | | |
| conductivity | conductivity | ● | ● | ● | | | | | | ● | | |
| salinity | salinity | ● | ● | ● | | | | | | ● | | |
| sea water turbidity | turbidity | | ● | | | | | | | ● | | |
| oxygen saturation | oxygen | | ● | | | | | | | ● | | |
| nitrogen saturation | nitrogen | | ● | | | | | | | ● | | |
| fluorescence | fluorescence | | ● | | | | | | | ● | | |
| chlorophyll A | chlorophyll_a | ● | | ● | | | | | | | | |
| sea water speed | curernt_speed | ● | | ● | | | | | | | | |
| sea water direction | current_dir | ● | | ● | | | | | | | | |
| Waves | | | | | | | | | | | | |
| wave mean period | wave_period | ● | | ● | | | | | | | | |
| wave maximum wavelength | wave_length | ● | | ● | | | | | | | | |
| wave significant height | wave_height | ● | | ● | | | | | | | | |
| wave direction | wave_dir | ● | | ● | | | | | | | | |
| Ship orientation | | | | | | | | | | | | |
| ship speed | ship_speed | ● | | ● | | | | | | | | |
| ship heading | ship_heading | ● | | ● | | | | | | | | |

| Variable | Name in data set | surface | profile | DShip | HATPRO | Ceilometer | DustTrak | Radiosondes | UAV | CTD | Calitoo | Microtops |
|----------|------------------|---------|---------|-------|--------|------------|----------|-------------|-----|-----|---------|-----------|
| ship heave | ship_heave | ● | | ● | | | | | | | | |
| ship heave std | ship_heave_std | ● | | ● | | | | | | | | |
| ship pitch | ship_pitch | ● | | ● | | | | | | | | |
| ship pitch std | ship_pitch_std | ● | | ● | | | | | | | | |
| ship roll | ship_roll | ● | | ● | | | | | | | | |
| ship roll std | ship_roll_std | ● | | ● | | | | | | | | |

## Appendix B: UAV flights

Here we give some more detailed information about the UAV operations, measurements and data post-processing.

**Measurement payloads and profiling strategy**

During ARC, four measurement payloads were used, each taking measurements during a single measurement flight. The primary focus of each package was different. Thus, different measurement strategies were employed during flights with specific packages. Below, a description of the measurement strategy for each package is provided.

1. Temperature, humidity and pressure profiling with the i-Met XQ2 package was conducted in the beginning of a CTD station. A continuous vertical profile from 5 m to 510 m ASL was conducted first with maximum vertical velocities (ascent rate of 4 m s$^{-1}$ and descent rate of 3 m s$^{-1}$). It was followed by a slower, tilted ascent-descent from 5 m to 110 m ASL, when the operator tried to maintain equal horizontal and vertical velocities of 1 m s$^{-1}$. The movement of the UAV was in an upwind direction to maintain aspiration of measurement sensors and limit the influence from the environment around the drone itself. At the end, another vertical profile from 5 m to 310 m ASL was conducted with the maximum ascent/descent velocities. It should be noted that if higher than normal battery consumption was noticed, the third profile was cancelled.

2. Horizontal wind speed profile with the Trisonica Mini package connected to the SKH1 logger (with temperature, humidity and pressure measurements). A single profile from 5 m to 510 m ASL during each flight was performed. A continuous vertical ascent with 5 m s$^{-1}$ velocity was followed by a descent with 40 s hoovers at predefined levels (typically: 510 m, 410 m, 310 m, 210 m, 110 m, 60 m, 27 m ASL). The continuous ascent was designed to collect temperature and humidity profiles, while hoovers during descent were designed to collect statistics for the horizontal wind observations. If the UAV battery showed abnormal consumption, an operator could omit some hoovers.

3. Aerosol (pm1, pm2d5, pm10) profile with the OPC-N3 package connected to the SKH1 logger (with temperature, humidity and pressure measurements). A single profile from 5 m to 410 m ASL during each flight was performed. A continuous vertical ascent with 5 m s$^{-1}$ velocity was followed by a descent with 90 s hoovers at predefined levels (typically: 410 m, 210 m, 110 m, 60 m, 18 m ASL). The continuous ascent was design to collect temperature and humidity profiles, while hoovers during descent were designed to collect statistics for aerosol observations. If the UAV battery showed abnormal consumption, an operator could omit some hoovers.

4. Upper ocean profiling with RBR package was conducted at least 100 m away from the vessel in the upwind direction in order to limit impact of mixing from the ship's propulsion. The profile was conducted by slowly (the operator tried to maintain a vertical velocity of less than 1 m s$^{-1}$) descending and ascending in and out of the water. The instrument was tethered at the end of 20 m long line in order to allow measurements down to 15 m while maintaining 5 m clearance between the UAV and ocean surface. Typically, three consecutive profiles were collected at the same location during a single flight. Flights with this package were only conducted in mild to moderate conditions that is when the sustained wind speed near the surface was below 6 m s$^{-1}$.

Ideally, during a single CTD station all four packages were employed. However, time constrains (e.g. shorter station) or environmental conditions (e.g. wind gusts) could have limited a number of flights. A typical sequence of packages deployed was: i-Met XQ2 – OPC-N3 – RBR – Trisonica Mini. Although this order sometime differed, all stations began with atmospheric measurements with i-Met XQ2 package.

**Data post-processing**

Data from each flight are processed as a single profile timestamped with its beginning (start_time), that is the time when UAV was at 5 m ASL at the beginning of measurements. Therefore, for example all data collected during a flight with the i-Met XQ2 package are used to calculate a single vertical profile of temperature and humidity. For continuous profiles of temperature, humidity and pressure from any sensor (i-Met XQ2, SKH, trisonica) only data from ascent legs of the flights are used. For horizontal wind speed (trisonica) and aerosol (OPC-N3) profiles only data from hoovers are used.

Post-processing includes correction of biases (constant offsets and linear drifts) calculated for each sensor/variable independently by comparison with data from the ship-based weatherstation. The first step is to calculate pressure offsets. These offsets were calculated during each cross-calibration period before and after each flight. It was determined that no sensor showed linear drift throughout the ARC measurement campaign and pressures of each sensor was corrected by a constant offset. These offsets (added to measured data) were: 1.07 hPa for i-Met XQ2 sensors, -2.21 hPa for Trisonica Mini sensor, 0.41 hPa for SKH1 sensor that was used during flights with the Trisonica Mini package and 0.57 hPa for SKH1 sensor that was used during flights with OPC-N3 package. Conversion between measured pressure and altitude was done using a constant rate of 8.5 m hPa$^{-1}$ and the known altitude of the helipad, that was 10.5 m ASL. That allowed temperature and humidity offsets to be calculated using collected data at the altitude of temperature/humidity observations by the ship-borne weatherstation (21.5 m ASL).

Humidity observations from all sensors were recorded as relative humidity. Any data points that showed saturation (relative humidity equal 100%) were removed. The i-Met XQ2 sensor is particularly prone to humidity drifts. As a first step in processing humidity data, specific humidity was calculated using recorded relative humidities as well as recorded temperature and pressure observations. In case i-Met XQ2, the humidity sensor had an auxiliary temperature sensor, which was used during conversion to specific humidity. All humidity offsets and corrections were then calculated using specific humidity. Next, relative humidity data were recalculated using bias-corrected specific humidity, bias-corrected temperature and bias-corrected pressure.

All sensors but the humidity sensors in i-Met XQ2 showed constant offsets throughout the ARC campaign. The i-Met XQ2 sensor showed a linear drift with time. Two regression lines were calculated for the two i-Met XQ2 sensors used: first for all profiles before Jan 31, 2023 and second for all profiles after that date. All offset values are provided as an attribute to each variable.

Data from hoovers (wind speed and particulate matter concentrations) were averaged at each available level. In case of wind data, the first 10 s of each hoover was removed from this calculation due to impact of rotor-induced turbulence on wind measurements. Wind data were also transformed from the recorded UAV-based frame of reference to the planetary frame of reference (zonal and meridional wind components) using compass information (pitch, roll and yaw) from the UAV flight computer.

**UAV setup and operations**

UAV operations carried out on a research vessel require a sufficiently large, safe and obstacle-free area for take-off and landing. During ARC, the helipad of the RV Maria S. Merian was selected due to its relatively large, even surface and unobstructed view to the operation side above the ocean. The typical procedure involved a fast vertical ascent and flying the UAV on the upwind side of the ship – a space where water was undisturbed by ship movements and direct positioning system operation. A typical horizontal range of operations was 50-100 m from the ship.

The fourth type of flight was an oceanic profile with an oceanic payload – RBR instrument attached to a 20 m long thin line. Special care was given to take-off and landing in this configuration. During the take-off, the payload operator held an instrument, while the UAV operator gently ascended and moved the vehicle to the windward side. The RBR instrument was gently let go to avoid unnecessary swing. During approach and landing these precautions were maintained as well. First, the payload operator caught the instrument and decreased the tension on the line to allow the UAV operator to safety land. A special care was given to swell to avoid any uneven strain on the line and the unexpected pull of the vehicle.

The landing procedure with either atmospheric or oceanic payload was initiated at a battery level of ∼30%. Landing was performed by catching the drone: as soon as the payload operator caught the drone, the UAV operator could switch it off immediately. The payload operator was wearing safety equipment, i.e. a helmet, eye protection and safety gloves.

*Author contributions.* JW and LK came up with the initial idea and the concept of this project. MB, LH, DK, SK, LK, BLa, BLo, TN, DP, PS, OT, and JW collected the data during the MSM 114/2 (ARC) cruise. DB, MC, LH, LK, BLo, TM, TN, DP, PS, AS, OT, and JW processed

the data. LK standardized the data sets developing the python package shipspy (Köhler, 2024) and documented the processing on Köhler

(2023). All authors wrote and edited the manuscript.

*Competing interests.* The authors declare that they have no competing interests.

*Acknowledgements.* In memory of Iris.

The authors would like to thank the crew of the RV Maria S. Merian for their great support. This work is supported by the Federal Ministry of Education and Research (BMBF) and the German Research Association (DFG) through funding for the research vessels. We would like
to thank Frank Nitsche for his support as chief scientist and his help with the CTD data.

LK, JW, LH, and DP thank Friedhelm Jansen for his help with setting up and dismounting the instruments, helping with more or less pressing problems and all the organisation around the campaign. LK wants to thank Hauke Schulz for his help with processing of the radiosonde data. LK and JW thank Tobias Kölling and Lukas Kluft for their help with the python package "shipspy" and their advice with respect to the data sets and processing. DB, MB, MC, DK, BL, and AS thank Aleksander Pietruczuk and Szymon Malinowski for their help
with processing the UAV data, and acknowledge funding from the Poland's National Science Centre (grants 2019/35/B/ST10/03463 and 2021/41/B/ST10/03660). LH, DP, and TM want to thank Bernhard Pospichal for his advice and help in processing the HATPRO data.

Thanks to the Maritime Aerosol Network (MAN) and in particular Alexander Smirnov for the permission to republish the Microtops data. The work by PS and OT was partially funded by EUMETSAT. LK was funded by the project DataWave within the Virtual Earth System Research Institute (VESRI) by Schmidt Futures founded by Wendy and Eric Schmidt.

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
