# Peer review of "Calm ocean, stormy sea: Atmospheric and oceanographic observations of the Atlantic during the ARC ship campaign"

_Earth System Science Data, 2024_

## Author Response (AR1)

**Response to Reviewer 1**

Dear Reviewer,

Thank you very much for your comments which helped to improve our manuscript a lot. In the copy of the manuscript with marked changes we highlighted the changes which we did as response to your comments in yellow. The changes coming from the comments of reviewer 2 are highlighted in green. General changes, mainly with respect to the data which are completely published now, are marked in blue. In the following, we respond to the specific comments one by one:

*Specific comments:*

*L34: What is SEABED? Maybe explain a bit or give a reference.*

We added an explanation and reference:
"The three main goals of ARC were to obtain vertically resolved cross sections of the ITCZ, to collect atmospheric and hydrographic reference data, in particular for comparisons with the Aeolus satellite and to contribute to the Seabed 2030 database, a global initiative to obtain a complete seabed map by the end of the decade (Mayer et al., 2018), and to examine the protist populations in the Atlantic along the route."
Mayer, L., Jakobsson, M., Allen, G., Dorschel, B., Falconer, R., Ferrini, V., Lamarche, G., Snaith, H., and Weatherall, P.: The Nip- pon Foundation—GEBCO Seabed 2030 Project: The Quest to See the World's Oceans Completely Mapped by 2030, Geosciences, 8, https://doi.org/10.3390/geosciences8020063, 2018.

*L62: "4,493 t" - Not sure about the guidelines for number separators, but the comma here looks a bit strange. Please check ESSD guidelines.*

We removed the comma.

*L66: Please add latitude and longitude of Mindelo*

We added "The RV Maria S. Merian left Mindelo located at 24.98°W 16.88°N on 22 January."

*L80: Please add latitude and longitude of Punta Arenas*

We added the coordinates for Punta Arenas:
"After leaving the Atlantic ITCZ, the RV Maria S. Merian headed for its port of disembarkation in Punta Arenas (70.91°W 53.15°S), Chile."

*L102: where on the ship the sea surface temperature is measured?*

The sensor is located in the echosounder equipment room. We added to line 102:
"Sea surface temperature is measured at 2 m depth by a PT100, 1/3 DIN B sensor with an accuracy of 0.1 K in the echosounder equipment room at the front tank deck starboard side."

*L109: "According to the manufacturer Sea-Bird, the sea surface temperature at"… here you mean sea water temperature, or?*

The definition of the depth of the sea surface temperature is not commonly agreed on. As pointed out in reference [1] "Inter-comparisons and combination of satellite and in situ measurements of SST are complicated by the fact that they are measuring different quantities". Satellites measure the first few mm as SST [1] whereas in situ observations typically take place between 10 cm and 10 m depth [2]. This can be solved by distinguishing skin SST from bulk or depth SST as proposed in [3,4].
We decided to call all temperatures measured close to the surface by sensors fixed to the ship sea surface temperature to distinguish them from the sea water temperature measured with the

CTD down to 3000 m depth. Thus, we refer to the temperature measured by the pure sea water system still as sea surface temperature.

*L128: "m/s" - please write it exponentially – m s-1 (check in the entire manuscript (for example L112, L137 and several others)*

We changed the notation of the units in the whole manuscript accordingly.

*L157: "TB" the subscript B should not be italic.*

We changed this in line 157 and the following paragraphs.

*L167: "10,000" I think a half space (\,) would be a better separator than a comma.*

Done.

*L183: Why the data quality cannot be guaranteed? Radome aging?*

MWR products could not be retrieved reliably during the storm event due to strong precipitation. Furthermore, there was a considerable amount of salt from the high waves in the storm on the HATPRO which deteriorated the data quality. For clarification, we added:
"We include data in Hayo et al. (2024) from the HATPRO until 15 February as data quality could not be guaranteed for later times times due to the storm event and connected strong precipitation and salt deposits on the instruments."

*L189: "the full backscatter profile is also stored but not integrated in this data set" – Where is it stored? Also on Pangaea? Please indicate that. - Ah you describe it later (L197). Maybe rephrase?*

We clarified this:
"For cloud and aerosol layer detection, the full backscatter profile is also stored in the level 0 raw data (Köhler et al., 2024f) but not integrated in the processed data set (Köhler et al., 2024e)."

*L260: "less than 4" - Is there a unit missing?*

No, the positional dilution of precision is a dimensionless quantity to estimate positional measurement precision. It uses the ratio between the variances of the different spatial components and the modelling error standard deviation [5].

*Table2: I recommend to put the caption on top of the table as it is more convenient.*

Done.

*L415: AOD is wavelength dependent. Please indicate the corresponding wavelength for AOD values.*

We added the wavelengths of Microtops and Calitoo:
"On 26 January a desert dust plume created elevated AOT values at 465 nm (Calitoo) and 440 nm (Microtops) of about 0.17, while on 29 January background AOT values of about 0.06 occurred."

*Fig3: Wind components in the caption must be italic (u, v)*

Done.

*L537: Typo? "pm2d5" should be pm2.5*

We note that pm2d5 is the name of the variable PM2.5 in our dataset.

References

[1] Embury, O., Merchant, C.J., Good, S.A. et al. Satellite-based time-series of sea-surface temperature since 1980 for climate applications. Sci Data 11, 326 (2024). https://doi.org/10.1038/s41597-024-03147-w

[2] Kent, E. C., and Coauthors, 2017: A Call for New Approaches to Quantifying Biases in Observations of Sea Surface Temperature. Bull. Amer. Meteor. Soc., 98, 1601–1616, https://doi.org/10.1175/BAMS-D-15-00251.1.

[3] Donlon, C. J., P. J. Minnett, C. Gentemann, T. J. Nightingale, I. J. Barton, B. Ward, and M. J. Murray, 2002: Toward Improved Validation of Satellite Sea Surface Skin Temperature Measurements for Climate Research. J. Climate, 15, 353–369, https://doi.org/10.1175/1520-0442(2002)015<0353:TIVOSS>2.0.CO;2.

[4] Wick, G. A., W. J. Emery, L. H. Kantha, and P. Schlüssel, 1996: The Behavior of the Bulk – Skin Sea Surface Temperature Difference under Varying Wind Speed and Heat Flux. J. Phys. Oceanogr., 26,1969-1988, https://doi.org/10.1175/1520-0485(1996)026<1969:TBOTBS>2.0.CO;2.

[5] Langley, R. B. Dilution of Precision. GPS World May 1999. http://gauss.gge.unb.ca/papers.pdf/gpsworld.may99.pdf

**Response to Reviewer 2**

Dear Reviewer,

Thank you very much for your constructive feedback. It has helped us to improve our paper, especially with regard to the HATPRO discussion. While we've tried to incorporate your suggestions, we've also tried to maintain a balanced perspective on all the instruments involved. In the copy of the manuscript with marked changes we highlighted the changes which we did as response to your comments in green. The changes coming from the comments of reviewer 1 are highlighted in yellow. General changes, mainly with respect to the data which are completely published now, are marked in blue. Detailed responses to your comments are set out below:

*General Comments*

- *the data set comprises different instruments from which the same variable is deducted given the overview in Tab 2, such as e.g. CWV retrieved from HATPRO, Microtops, and radiosoundings; or thermodynamic profiles obtained from UAV, soundings and HATPRO. The manuscript in its current form lacks quantification and discussion of how these measurements compare to each other. This would additionally information which one of the available measurements is best to use; if biases are present between instruments and how high they are (especially interesting regarding the soundings, e.g. see Stephan et al, 2020); and how accurate the measurements are. An additional (sub)-Section should be added, maybe within Section 4.*

This is a valid point and we have now included a brief comparison in section 5.1. However, we believe that a comprehensive cross-evaluation of all instruments is beyond the scope and purpose of this paper. Rather, the idea of the paper is to present the data in a format that facilitates comparison. We therefore limit our analysis to the one example in section 5.1. We have also included a statement to encourage readers to do something comparable (where possible) for their specific purposes by adding to the introduction: "As this challenge is not unique to the ARC campaign, the purpose of this paper is twofold. First, to publish a coherent dataset of the data collected during the campaign which facilitates studying research questions using different variables from different instruments as well as comparisons of the same quantity measured by different instruments with different techniques. Second, to provide the necessary processing steps and scripts to serve as a prototype for future (and past) ship campaigns.".

Section 5.1. now includes the column-integrated water vapour obtained from the radiosonde ascents as well as from the Microtops in the upper panels of the three crossings in Figure 4 such that it can be directly compared to the HATPRO CWV. We find a nice agreement between the HATPRO and the radiosonde CWV, although the radiosondes show slightly larger values due to the fact that they integrate over the whole troposphere whereas the HATPRO only includes the troposphere up to 10 km. The Microtops also mostly coincides with the other two instruments. However, in the second part of crossing 3 it shows enhanced values possibly due to undetected clouds, as the CWV determination with the Microtops only works with clear sky.

We introduced the CWV values of the radiosondes and Microtops in the second paragraph of section 5.1 as follows:

„The blue stars correspond to the CWV derived from radiosonde ascents by integrating the water content. To make sure that the integration starts at sea level, we restrict ourselves to ascents. The blue crosses show the column-integrated water vapour from the Microtops observations."

We added the comparison and interpretation of the CWV observations at the end of this section:

"Comparing the CWV from the HATPRO with the radiosondes during the crossings, we find that both instruments are in good agreement with a root mean square deviation (RMSD) of 4.3 mm. The CWV from the radiosondes, however, tends to be slightly higher which is likely due to the fact that it includes the whole troposphere whereas the HATPRO only integrates up to 10 km height. Furthermore, the radiosondes are not affected as much as the HATPRO by precipitation events and thus do not show excessively large signals during rain. The Microtops uses the 940 nm absorption line to determine the column water vapour in clear sky. In crossing 1 and 2 and at the beginning of crossing 3, it coincides closely with the other two instruments. The RMSD of the Microtops CWV from the HATPRO CWV until 2 February is 2.7 mm. However, in the course of crossing 3 from 3 February on, the Microtops shows very high CWV values and a RMSD from the HATPRO of 19.2 mm. This, together with the precipitation rate as indicated by the HATPRO and

the cloud base height values detected with the ceilometer, suggests that the conditions were not completely cloud free despite the post processing and shows the high sensitivity of the Microtops to the presence of clouds. We therefore show the CWV values from the Microtops after 3 February only with light blue crosses. In summary, we find that the HATPRO CWV coincides very well especially with the radiosoundings but also most of the time with the Microtops. This comparison illustrates more generally how the availability of data in a consistent way not only facilitates the comparison of different quantities, but also the comparison of measurements of the same quantity made by different instruments to assess the quality of the data."

- *Retrieval information content of temperature and absolute humidity retrieved from HATPRO is generally highest below 4km (e.g. Crewell and Löhnert, 2007; Löhnert and Maier, 2012). Löhnert and Maier (2012) find that 'only 5 % independent information originates from the radiometer measurement itself' above this height, which makes the derived profiles quite insensitive to the actual atmospheric conditions. A comparison between HATPRO and soundings should be added to illustrate the strengths and limitations of the provided HATPRO-derived profiles, as these profiles seem to be a more central part of the atmospheric part of the data set presented in Sec 5.*

Thank you for raising this important point. In order to emphasise the height dependency of the data quality of the HATPRO measurements, we have made two changes to the paper. First, we have added the references suggested by the reviewer to provide more detailed information about the HATPRO profiles, as well as a discussion of the limitations in the upper free troposphere in Section 3.2, see also below in the respective responses to the specific comments. Secondly, we have added a simple comparison between radiosonde and HATPRO measured relative humidity, split into the height range above and below four kilometres. We calculated the root mean square deviation of the relative humidity measured by the radiosondes and the HATPRO. As expected from the references listed by the reviewer, there is a much better agreement between the humidity in the lower free troposphere (RMSD = 14%) than in the upper free troposphere (RMSD = 22%). We added this comparison to section 5.3:
"Due to the decreasing accuracy of the HATPRO with height (see section 3.2), the quantitative HATPRO profiles are more reliable below four km than above. This is also reflected when comparing the HATPRO and radiosonde relative humidity by calculating the RMSD. Below four km, we find it to be 0.13 whereas in the range between 4 and 10 km it is 0.22."

*Specific Comments*

- *a short review on past ship campaigns in the same region should be added to the introduction, including how the ARC measurements complement those previous statistics.*

We have added the following paragraph to the introduction, referring to past and future ship campaigns in the tropical Atlantic with very similar data sets and main objectives, but in different seasons and at different longitudes.
"ARC took place during boreal winter and therefore complements the Mooring Rescue ship campaign in summer 2021 with the RV Sonne, reference SO284 (Brandt et al., 2021), and the BOW-TIE ship campaign with the RV Meteor, reference M203, in summer 2024 (Leitstelle Deutsche Forschungsschiffe, 2024a). Although they were conducted at different times of the year and investigated different parts of the tropical Atlantic than the ARC campaign, a common core objective of the campaigns was to create vertical profiles of the ITCZ through the thermocline to the tropopause. An additional ship campaigns with a similar suite of instruments is planned for the beginning of 2025 (M207 with RV Meteor (Leitstelle Deutsche Forschungsschiffe, 2024b)). Processing the data from the various campaigns in the same way as for ARC will greatly facilitate the study of seasonal and regional changes in the Atlantic ITCZ."

- *Fig 1: to improve readability, the inlet in panel (a) could be enlarged to highlight the three crossings. Color-coding the track with the corresponding date in one of the panels would help the reader in the following to associate the given examples with the ship's location.*

We increased the size of the figure by using two rows. Furthermore, we color coded the time in the top inset of figure 1(a) and added the following information to the main text:

"The track in the inset is thus colour coded indicating the time starting with blue on 26 January via red to green on 5 February."
The figure caption was adjusted as well:
„(a) Route with zooms in the insets. The colour coding in the upper inset refers to the time from blue (26 Jan) via red to green (5 Feb). The red dashed line indicates the position of the storm."

- *Fig 2: the ship track could be added e.g. in a transparent grey to illustrate the dimensions of the cloud systems versus ship movements*

We added the ship track as bright yellow line and added the information to the figure caption.
"The yellow line indicates the ship track for reference."

- *Sec 3: it would be nice to add a photo or schematics to illustrate where exactly and how far apart the instruments were located aboard the ship.*

We included a picture of the RV Maria S. Merian taken with the UAV and marked the instrument positions in the new Figure 3.

- *Sec 3: I understand that none of the instruments were mounted on a stabilizing platform. A clarifying statement on if and how measurements were affected by ship motion should be added, and if the provided data sets were corrected or flagged for ship motion.*

Many of the instruments on board were not sensitive to the ship's motion. However, the HATPRO and Ceilometer are affected because they are fixed to the ship. Especially in the tropics, the sea was very calm, so we think the error is small except for the storm, for which the HATPRO data is already not included and the Ceilometer data is also flagged because the window was full of salt. We have included the following statement concerning the ship's motion to the beginning of section 3:
"Most instruments were either not sensitive to the ship's motion or they were adapted to it in various ways. The radiation instruments on the RV Maria S. Merian are mounted on a two-axes gimbal for horizontal stability. The sunphotometers were handheld and thus adapted to the ship's motion by manually pointing to the sun. The DustTrak was not sensitive to the ship's motion because it pumps ambient air. The Ceilometer and the HATPRO were fixed on the observation deck, so they moved with the ship. However, we do not consider this a significant error since the angular deviation was small ($< 6.2°$) except during the storm ($< 22.2°$) where also strong precipitation and salt water from large waves were deteriorating the measurements. The information about the ship's motion is given in the DShip data set parameters ship_roll and ship_pitch."

- *Tab 1: Which generation of HATPRO was used?*

A Generation 5 (G5) HATPRO was used in this campaign. We added this information to the table.

- *Sec 3.1: I am curious how the logged times were matched amongst instruments: were they synchronised with the Dship system, or was a post-processing applied to the data for time stamp matching? It would be nice to add a sentence here to clarify.*

Most of the instruments were connected to the ship's network and thus the time stamp was automatically synchronised with DShip. UAVs, Microtops, and Calitoo used GPS time, such that the time was in agreement with DShip as well. For the DustTrak, the time stamp was synchronised with the ship time at the beginning of each measurement and checked in the post-processing. We added an explanation to the beginning of section 3:
"Most of the instruments were connected to the ship network and thus their time stamps were automatically synchronised with DShip or used GPS timestamps which were in agreement with DShip. For the DustTrak, the time was manually synchronised with DShip on a regular basis. All time stamps were checked during the post processing."

- *L 172: LWP error scales with absolute value of LWP (Jacob et al, 2019; Schnitt et al, 2024). Does the range of measured LWP values correspond to the uncertainty range given here?*

The LWP uncertainty estimate is updated, taking into account the increase in absolute error with rising LWP values. We updated the text as follows:
"Retrieval error of CWV is estimated around 0.5-0.8 kg m−2 (Steinke et al., 2015), and for LWP the relative uncertainty ranges from over 100% for LWP below 15 g m−2, to 50% for LWP around 40 g m−2, and decreases to 20% for LWP above 100 g m−2 (Jacob et al., 2019)."

- *L 173: the stated uncertainties of temperature and humidity profiles from Löhnert et al (2009) are derived based on simulated clear-sky cases using an optimal estimation retrieval combining HATPRO and infrared measurements which is a different retrieval approach than the one described here. Given the presented stand-alone HATPRO measurements in all-sky conditions derived from a statistical retrieval, the uncertainties should rather be estimated by comparing HATPRO-derived temperature and humidity profiles with e.g. sounding profiles, such as performed e.g. in Löhnert and Maier (2012); Walbröl et al, 2022 (also see General Comment 1 and 2). An additional sentence should be added describing the information content of obtained profiles with height. How do these uncertainties translate into the calculated relative humidity and fields?*

Error estimates and references for the profile information are updated to reflect the characteristics of the applied retrieval method. The information content and relative humidity error estimate are now mentioned as well.
"Profiles of atmospheric temperature and humidity can generally be retrieved with errors below 2 K (Löhnert and Maier, 2012) and less than 1 g m−3 (Walbröl et al., 2022), respectively, when compared to radiosondes in the lowest 4 km. This translates to an error of up to 0.2 in relative humidity. The information content decreases with height, with about 10 % and 20 % of the temperature and humidity information, respectively, coming from heights above 500 hPa (Ebell et al., 2013). Higher altitudes are therefore mostly influenced by the mean state given in the training data set."

- *L 181: I understand that averaging the LWP product to 1-minute is chosen to find a compromise between different temporal resolutions of all instruments (L 338). LWP often varies strongly over the course of 1 minute in the Tropical Atlantic conditions. It would be great to additionally include the higher-resolution data as an extra data set, including the TB measurements; and to include the LWP standard deviation as an additional measure to the data set Hayo et al, 2024. Additional information on whether and how a clear-sky correction was applied to the retrieval should be added. Was the ceilometer used?*

An offset correction was applied to the retrieved LWP using a 2 min brightness temperature standard deviation of the MWR window channel at 31.4 GHz. Liquid water cloud free scenes are detected within a 20 min window using a threshold of 0.1 K times the median ratio of the water vapour (22.24 GHz) and window channel to account for a water vapour dependency of the threshold. Offset values are also stored in the data set (lwp_offset). Adding the ceilometer wouldn't necessarily improve the method, since it would introduce additional uncertainties due to time matching and different measurement volumes. We added the following information:
  "With the application of a statistical LWP retrieval, nonphysical bias values can occur and be detected during clear sky cases. Therefore, an offset correction was applied to the retrieved LWP using a 2 min brightness temperature standard deviation of the MWR window channel at 31.4 GHz. Liquid water cloud free scenes are detected within a 20 min window using a threshold of 0.1 K times the median ratio of the water vapour (22.24 GHz) and window channel to account for a water vapour dependency of the threshold. Offset values are also stored in the data set (lwp_offset, Table 2)."

- *L 309: Why only on the ascents?*

The reason for this is to avoid contamination from the UAV itself. We included the following explanation at the end of the paragraph introducing the UAV payloads:
„The atmospheric payloads (1-3) were placed on top of the UAV to avoid contamination from its dark frame and the radiators located in its lower part. Thus measurements during

ascents are collected in an undisturbed environment while measurements during descents may be contaminated by the diabatic heating from the UAV frame. Hence, only ascent data are used in the atmospheric profiling."

- *Tab 2: I like having an overview of all measured parameters included in the manuscript. In its current version, I find the table a little hard to navigate when looking for a specific variable (especially when printed). I would propose to either sort alphabetically; or include sub-titles to sort the variables (e.g. navigation; clouds; aerosols; …) As the table covers a few pages, I propose to move the Table to an appendix in order to facilitate navigating the manuscript; and to include a shorter version here for overview purposes.*

We included titles for the different variable groups which we split in time and position, meteorology, aerosols, sea water, waves, clouds, and ship orientation. We furthermore reduced the table by skipping error and standard deviation parameters as well as technical and some derived variables. However, we didn't reduce the table further because we want to avoid biasing the reader in the choice of variables and subjects to study with the data sets. The full table is shown in the appendix as suggested by the reviewer.

- *L 423: the HATPRO-derived profiles contain almost no information content at this height (see General Comment #2). I would suggest to rather refer to the soundings' relative humidity fields (also Fig 3b) RH panel), and add a comparison between HATPRO and soundings profiles to illustrate RMSEs and biases between the obtained profiles (and, thus, sensitivity of the HATPRO)*

We now discuss the dependence of the HATPRO data on height in more detail in section 3.2.
With this background information, we hope that the reader will be able to interpret the profiles correctly and acknowledge that the profiles become less accurate with height.
In this figure, we want to give an overview of all atmospheric instruments included in the data set. Therefore, we don't want to show the relative humidity from the radiosoundings twice but rather also include the HATPRO profiles. We also think, showing the relative humidity is useful for the interpretation of the structure of the ITCZ during this second crossing since it nicely shows the drier doldrum regions and the moister edges and can also be connected with the cloud base height from the Ceilometer. Therefore, we keep the lowest panel of Figure 3b (new 4b) as before but remind the reader of the lower accuracy with height in the interpretation of the Figure:
"Note that the HATPRO profiles are more accurate below 4 km than above as discussed in Section 3.2."

- *L 426: did cloud base really reach ship level at 0km, or was there precipitation present influencing the derived cbh?*

The minimum cloud base was 322 m, and this is also the order of magnitude of the minimum values in the lowest panel of Figure 3b (new 4b). Although it might not seem like this due to the altitude range shown in the plot, the minima of the cbh are different but around a few hundred meters. Therefore, there was no fog but very low clouds. This coincides with our impression onboard during the campaign. We specified the cloud base height range more precisely:
"In the convective regions, the cloud base height was between 322 m and 10 km"

- *L 463: Adding the wind speed here as a panel in Fig 4 would underline this statement.*

In Fig 4 (now Fig 5), we want to focus on the comparison of the profiles from different instruments of the same quantity. Furthermore, we already showed the surface wind speed for all three crossings in the previous figure. To avoid duplication, we decided to leave Figure 4 (new Fig 5) limited to relative humidity plots but reformulated the statement in line 463 as:
"Comparing the wind speeds plotted in Figure 4 with the relative humidity shown in Figure 5, we find that, as noted above and illustrated in Figure 4(c), low wind speed events tended to coincide with reduced humidity, especially in the free troposphere."

*Technical Corrections*

- *Fig 1: to improve readibility, axes labels should be increased.*

Done.

- *L 470 reads oddly; I recommend combining the sentences.*

We changed the formulation to:
"The bottom panel (d) shows CTD profiles of the ocean down to 500 m which is the maximum depth of the CTD scans with two exceptions as mentioned above. On most days, two CTDs were deployed."

- *a reference to the cruise report should be added*

We included the reference of the short cruise report to the introduction.

- *update of reference: Schnitt et al, 2024 was published*

Done.